# TabWak: A Watermark for Tabular Diffusion Models

**Chaoyi Zhu**[1]**, Jiayi Tang**[1]**, Jeroen Galjaard**[1]**,**
**Pin-Yu Chen**[3]**, Robert Birke**[4]**, Cornelis Bos**[1,5]**, Lydia Y. Chen**[1,2]

TU Delft[1], University of Neuchâtel[2], IBM Research[3], University of Turin[4], Tata Steel Research[5]
`{c.zhu-2, j.m.galjaard, c.bos, y.chen-10}@tudelft.nl`
`j.tang-14@student.tudelft.nl`
`pin-yu.chen@ibm.com`
`robert.birke@unito.it`

## ABSTRACT

Synthetic data offers alternatives for data augmentation and sharing. Till date, it remains unknown how to use watermarking techniques to trace and audit synthetic tables generated by tabular diffusion models to mitigate potential misuses. In this paper, we design `TabWak`, the first watermarking method to embed invisible signatures that control the sampling of Gaussian latent codes used to synthesize table rows via the diffusion backbone. `TabWak` has two key features. Different from existing image watermarking techniques, `TabWak` uses self-cloning and shuffling to embed the secret key in positional information of random seeds that control the Gaussian latents, allowing to use different seeds at each row for high inter-row diversity and enabling row-wise detectability. To further boost the robustness of watermark detection against post-editing attacks, `TabWak` uses a valid-bit mechanism that focuses on the tail of the latent code distribution for superior noise resilience. We provide theoretical guarantees on the row diversity and effectiveness of detectability. We evaluate `TabWak` on five datasets against baselines to show that the quality of watermarked tables remains nearly indistinguishable from non-watermarked tables while achieving high detectability in the presence of strong post-editing attacks, with a 100% true positive rate at a 0.1% false positive rate on synthetic tables with fewer than 300 rows. Our code is available at the following repository https://github.com/chaoyitud/TabWak.

## 1 INTRODUCTION

Synthetic data from generative models is becoming integral to today's data management and artificial intelligence services. Synthetic tables generated from tabular generative adversarial networks (Zhao et al., 2021; Xu et al., 2019) and tabular diffusion models (Kotelnikov et al., 2023) are used to augment the data for training machine learning models and substitute the original data for protecting privacy (Guo & Chen, 2024). Synthetic tabular is the most common modality in industry and organizations, which increasingly embrace synthetic data as a privacy-preserving data-sharing solution (Liu et al., 2022; Qian et al., 2024; Potluru et al., 2024). It is important for the synthetic data generator to verify if a piece of table is generated by itself and then take responsibility for the (misa)usage of such data. Synthetic tables pose subtler yet significant risks. For instance: 1) Financial Fraud: Synthetic datasets can manipulate performance metrics, enabling hedge funds to fabricate high returns and conceal losses. Watermarking ensures that only genuine data is used for informed decision-making. 2) Healthcare Misdiagnosis: Altered synthetic patient data can skew diagnostic tools or treatment recommendations, potentially leading to issues like over-prescription of medications. Watermarking safeguards data integrity, fostering trust in healthcare models. 3) Regulatory Evasion: Companies may exploit synthetic data to falsify compliance records, inflate profits, or create misleading sustainability reports. As such, synthetic data is increasingly adopted

for critical tasks, it is paramount to ensure its traceability and auditability to avoid harm and mis-usages. Recent advancements in watermarking technology (Kirchenbauer et al., 2023; Kuditipudi et al., 2023; Wen et al., 2023; Zhu et al., 2024; Yang et al., 2024) have demonstrated significant promise in texts from language models and images from diffusion models. The key challenges of designing watermarks are twofold: the trade-off between the data quality and detectability, and their robustness against post-editing operations, such as deletions and insertions (Kuditipudi et al., 2023).

Existing studies on image and language generative models focus on embedding watermarking keys during the training (Fernandez et al., 2023), sampling (Wen et al., 2023; Kirchenbauer et al., 2023), and post-editing phases (Topkara et al., 2006; Barni et al., 2001; He et al., 2024). Sampling-phase watermarking, which alters only the sampling process without changing the model weights yet maintains high data quality (Wen et al., 2023; Kirchenbauer et al., 2023), offers a favorable trade-off between computational overhead and robustness. In the context of token-based large language models (e.g., GPT), secret keys (Kirchenbauer et al., 2023) are used to modify the logit values of vocabulary tokens, thereby adjusting token probabilities for the next-word generation according to the context and keys. For image diffusion models, watermarking is proposed to be embedded in the latent space (Wen et al., 2023; Yang et al., 2024). Despite substantial research on watermarking synthetic texts and images, there is, unfortunately **no study on watermarking tabular generative models during the sampling phase**.

Existing techniques for watermarking diffusion models (Wen et al., 2023; Yang et al., 2024) achieve a good balance between data quality and detectability on images. However, they do not allow for direction application to the tabular domain. Applied at table level, such watermarks become susceptible to common row-level operations like sorting, shuffling, and selection, which hinders detectability. Conversely, applied at the row level using a fixed pattern across rows for row-order independence diminishes cross-row diversity, ultimately negatively impacting the quality of the generated data. This challenge between ensuring watermark robustness and data diversity underscores the need for a new approach that safeguards against row-level transformations while preserving the overall quality and diversity of the generated tabular data.

In this paper, we propose the first watermarking scheme, `TabWak`, for tabular generative models in the sampling phase. Particularly, we consider a Latent Diffusion Model (LDM) (Zhang et al., 2024) that encodes heterogeneous (i.e., both continuous and categorical) variables into a unified latent space via auto-encoder networks on which diffusion models synthesize latent codes. In `TabWak`, we preserve the latent distribution to be close to the model's assumptions (i.e., the standard Gaussian) while enabling row-wise detection for tabular data. During synthesis, a joint self-cloning and seeded shuffling technique ensures row-level variation, preventing repetitive patterns to prevent synthetic table quality degradation due to repetition. During detection, our proposed valid bit mechanism increases robustness against distortions and attacks, with theoretically guaranteed improvements in detection reliability.

We evaluate `TabWak` on five datasets with synthetic tables generated by TabSyn (Zhang et al., 2024) under normal and adversarial post-editing settings. Therein comparing the data quality, detectability, and robustness of `TabWak` against two state-of-the-art baselines, Tree-ring (Wen et al., 2023) (TR), and Gaussian Shading (Yang et al., 2024) (GS). Due to its close alignment with the standard Gaussian distribution and row-level latent variation, `TabWak` imposes minimal loss of data quality over original synthetic data in both terms of in terms of shape, trend, discriminability, and machine learning performance (MLE). Moreover, as measured by Z-scores (Casella & Berger, 2024) across all five datasets, `TabWak` demonstrates strong detectability of detecting watermarks with only 1K rows when the Z-score is higher than 3.95, which corresponds to a theoretical false positive rate below $3.9 \times 10^{-5}$. To assess the robustness of `TabWak`, we designed five post-editing attacks: deletion at the row, column, and cell level, plus noise injection and shuffling across rows. Our row-wise detection mechanism ensures inherent robustness against row-level attacks, such as row deletion and shuffling, without any loss in detectability. Furthermore, our valid bit mechanism provides enhanced resilience against other forms of attack, ensuring best robustness among all methods.

Our primary contribution is the design and validation of `TabWak`, the pioneering watermarking scheme for latent tabular diffusion models. The technical contributions are detailed as follows:

- We propose a novel watermarking technique that enables row-wise embedding in tabular data with minimal impact on data quality over non-watermarked results regarding statistical and machine learning performance.

- We introduce a valid bit mechanism to enhance the watermark's detectability under adversarial post-editing attacks.

- We derive a theoretical guarantee regarding row-level diversity and detection effectiveness.

- We develop a comprehensive benchmark for evaluating the robustness of tabular watermarks, incorporating five distinct attack types targeted at tabular data.

- Extensive empirical evaluation demonstrating that `TabWak` meets three key objectives: i) preserving the quality of synthetic data, ii) achieving high watermark detectability across multiple datasets, with average Z-score significantly exceeding $3.95$, and iii) ensuring robustness, with $100\%$ true positive rate at a $0.1\%$ false positive rate on synthetic tables with fewer than 300 rows under various post-editing attacks.

## 2 RELATED WORKS

**Watermarking Synthetic Data**   With the ability to create contents that mimic human creativity, generative AI models have achieved notable proficiency in generating high-fidelity images, videos, texts, and more (Borsos et al., 2023; OpenAI, 2023). However, this progress also introduces challenges, notably the potential for misuse, such as deepfakes and misinformation enabling fraud and scams (Schreyer et al., 2019; Gupta et al., 2023; Karnouskos, 2020). To ensure accountability against potential misuse and risks, watermarking across various data modalities has been proposed as an effective strategy to enhance traceability by embedding hidden signatures into all generated content.

**Watermarking Images and Text**   Watermarking can be integrated into generative models by modifying training procedures through explicit training or modified sampling. The former involves embedding a watermark into the training data, ensuring that the generated images (Yu et al., 2021a;b; Zhao et al., 2023) or text (Tang et al., 2023; Sun et al., 2023) inherently contain the watermark. The latter, on the other hand, does not require the re-training of a generator per watermark. For images Pivotal Tuning Watermarking (Lukas & Kerschbaum, 2023) offers a method for watermarking pretrained GANs by adjusting the models during post-training. Other methods use the invertibility of diffusion models (Wen et al., 2023) or employ additional encoders and decoders to embed a watermark message-matrix (Xiong et al., 2023). without retraining. For text, alternatives use the token sequence to modify the probability distribution of the next predicted token either during the logits generation (Kirchenbauer et al., 2023; Zhu et al., 2024) or directly during the token sampling phase without modifying the logits. The latter can be implemented at the word (Kuditipudi et al., 2023) or sentence (Hou et al., 2023) level. Yet, due to their assumption on representation order, these methods fail to address the need for diversity at scale and resilience to column reordering of tabular data.

**Watermarking Tables**   Recent works (He et al., 2024; Zheng et al., 2024) on watermarking synthetic tabular data have focused on embedding watermarks through additive post-editing noise to ensure numerical values fall into strategically chosen intervals. However, no existing method addresses watermarking at the sampling phase, where the watermark is embedded in the noisy latent space rather than directly modifying the tabular data itself. In this work, we extend watermarking techniques to latent tabular diffusion models in the sampling phase with our proposed `TabWak`, which maintains high synthetic data quality, achieves superior watermark detectability, and demonstrates strong resilience against post-editing attacks.

## 3 TABWAK: ROW-WISE TABLE WATERMARKING

The primary distinction between diffusion models for images and tables lies in the nature of their latent representations. For image diffusion, the latent representation encapsulates all pixels of a single image as a unified whole, allowing watermarking techniques to target this holistic representation. In contrast, tabular diffusion models generate row-specific latent representations, where each row of a table is treated as an independent unit to enhance the diversity between rows and watermark robustness to post-editing attacks. This is analogous to watermarking a batch of independent

images, where each row behaves like a separate image. These fundamental differences introduce additional challenges that make a straightforward application of image-based watermarking methods unsuitable for tabular diffusion models, demarcated by three key factors:

- Independent Row Units: Row-level operations such as row shuffling, deletion, or reordering are common when handling tables. This necessitates row-wise detection of watermarks rather than treating the table as a unified entity. The watermarking technique proposed by (Wen et al., 2023), for example, embeds the watermark in the Fourier space of the latent space, treating the table holistically. This approach, however, is unsuitable for row-by-row detection in tabular data, where the dependence across rows is crucial for detecting any alterations or attacks at a granular level.

- Latent Representation Diversity: Unlike images, where different prompts naturally contribute to diverse generated outputs, tabular data lacks this diversity enhancer. In fact, if the latent representations across rows are too homogeneous, e.g., due to equal per-row watermarking like Yang et al. (2024), the quality and utility of the generated table can significantly degrade. Therefore, ensuring diversity across row-wise latent representations is critical for maintaining the table's integrity while embedding a robust watermark.

- Unique Post-editing Attacks: Post-processing in the tabular domain differs significantly from the image domain. Tabular-specific attacks, such as row deletion, row shuffling, and column deletion, require a watermarking scheme to be robust against these unique challenges.

In summary, our watermarking method focuses on the following: (i) Performing watermarking row-wise, thereby alleviating the dependence on row-ordering during detection. (ii) Ensuring row-wise diversity in the latent representations via self-cloning with shuffling. (iii) Robustness against post-editing attacks.

**Latent Diffusion Models Inversion** Latent diffusion models use latent variables $z_t$ for $t \in [T, ...0]$ throughout the generation process, necessitating a decoder $\mathcal{D}$ that converts the denoised latent variable $z_0$ into tabular data $X_0$. Watermark detection necessitates reversing the process. During detection, we reconstruct the latent tabular data through iterative gradient descent, following (Hong et al., 2024). Thereby reducing latent reconstruction error due to the in-exact mapping of the encoder and decoder mapping. Furthermore, to recover the latent variable $z_T$, we incorporate the DDIM (Denoising Diffusion Implicit Models) inversion (Wen et al., 2023), enabling the recovery of $\hat{z}_T$ from the reconstructed latent $\hat{z}_0$ following the decoder inversion step.

## 3.1 GAUSSIAN WATERMARK EMBEDDING

The dimension of the latent representation $z_T$ for a single row is defined as $m$. Let $\phi(x)$ denote the probability density function of the standard Gaussian distribution $\mathcal{N}(0, 1)$, and $\Phi(x)$ represent its cumulative distribution function (CDF). The function $\phi(x)$ is partitioned into $l$ quantiles (segments) of equal cumulative probability. Using the diffusion-preserving approach in (Yang et al., 2024), we construct a control seed $d$ consisting of integers of the length $m$, where each element $d_i \in [0, l)$. When $d_i = k$, the watermarked latent representation $z_T^w$ is constrained to the $k$-th quantile of $\phi(x)$, implying that $z_{T,i}^w$ is sampled from the following conditional distribution:

$$p\left(z_{T,i}^w \mid d_i = k\right) = \begin{cases} l \cdot \phi\left(z_T^w\right) & \text{if } \Phi\left(\frac{k}{l}\right) < z_{T,i}^w \leq \Phi\left(\frac{k+1}{l}\right) \\ 0 & \text{otherwise.} \end{cases}$$

**Self-Cloning plus Shuffling Mechanism** Previous methods for image generation (Yang et al., 2024) use $d$ as the control seed. Using the same $d$ for all rows leads to poor diversity (see Appendix F for an example). Conversely, using row-specific $d$ requires matching each row to the correct $d$ during detection, which makes the watermark vulnerable to even simple row reordering attacks. In contrast, we introduce a self-cloning plus shuffling mechanism that embeds the secret watermark key into the ordering of the elements in $d$. This allows the generation of distinct control seeds while ensuring row-level detectability via a unique watermark key.

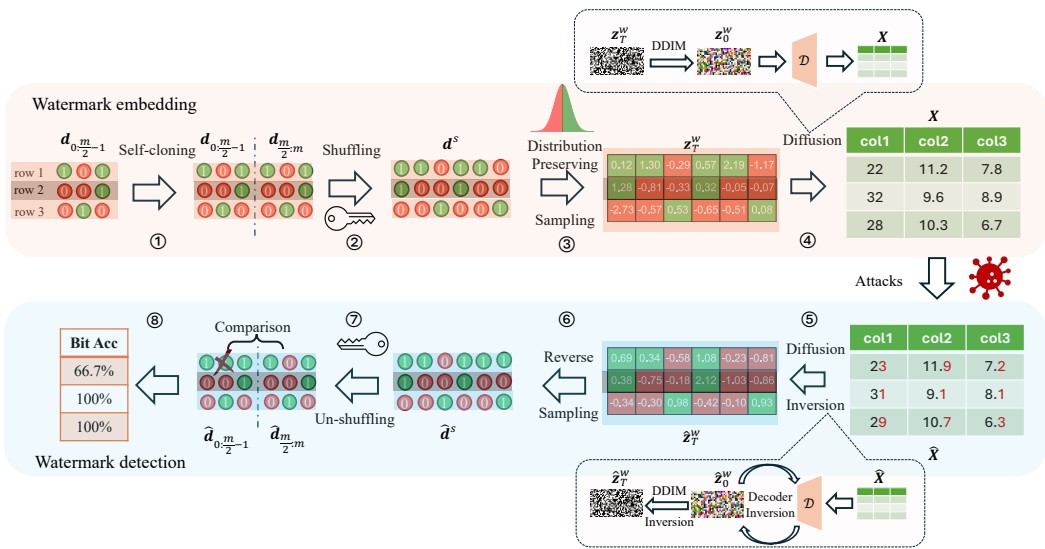

Figure 1: The framework of TabWak. The first half of the control seed, $d_{0:\frac{m}{2}-1}$, is randomly drawn from a discrete uniform distribution over $0, 1$. After self-cloning and shuffling, the control seed is used for distribution-preserving sampling. The tabular data is then generated by the latent diffusion model. During detection, the control seed is recovered through diffusion inversion, reverse sampling, and un-shuffling. Finally, the bit accuracy between the first and second halves of each row is checked.

The proposed method divides the control seed $d$, with length $m$, into two parts: $d_{0:m/2-1}$ and $d_{m/2:m}$, where $m$ represents the dimensionality of the latent vector for a row. In our experiments, $m$ is a model- and data-related hyperparameter, calculated as the product of the token dimension and the number of columns in the table. The first part, $d_{0:m/2-1}$, is sampled from a discrete uniform distribution over the set $\{0, 1, \ldots, l-1\}$, while the second part, $d_{m/2:m}$, is set to be identical to $d_{0:m/2-1}$. The elements of the sequence $d$ are shuffled using a pseudo-random permutation, seeded by the watermark key $\kappa$, to produce the row signature $d^s$. Let $d = (d_0, d_1, \ldots, d_{m-1})$ represent the control sequence. The shuffling process can be described using a permutation $\pi_\kappa$ of the indices $\{0, 1, \ldots, m-1\}$, where $\pi_\kappa$ is determined by the watermark key $\kappa$.

The new shuffled signature $d^s$ is then given by:

$$d_i^s = d_{\pi_\kappa(i)} \quad \text{for } i = 0, 1, \ldots, m-1.$$

Given $u \sim U(0, 1)$ from a discrete uniform distribution, we subsequently sample the latent variable $z_T$ as:

$$z_T^w = \Phi^{-1}\left(\frac{u + d^s}{l}\right)$$

where $\Phi^{-1}(\cdot)$ is the percent point function (PPF) of a standard Gaussian distribution $\Phi(\cdot)$.

For detection and extraction of the watermark, the inverse mapping is given by,

$$d^s = \lfloor l \cdot \Phi(z_T^w) \rfloor.$$

Given the watermark key $\kappa$, the original control seed $d$ can be recovered from the shuffled sequence $d^s$ using $\pi_\kappa^{-1}$, which reverses the effect of the shuffle $\pi_\kappa$. The inverse-shuffled sequence it then obtained following,

$$d_i = d_{\pi_\kappa^{-1}(i)}^s \quad \text{for } i = 0, 1, \ldots, m-1,$$

where $\pi_\kappa^{-1}$ is the inverse-permutation function mapping each permuted each index $i$ in $d^\kappa$ back to its original index in $d$.

The bit accuracy is then defined as,

$$A_{bit} = \frac{1}{m/2} \cdot \sum_{i=0}^{m/2-1} \mathbb{I}(\boldsymbol{d}_i = \boldsymbol{d}_{m/2+i}),$$

where $\mathbb{I}(\cdot)$ is the indicator function returning 1 if the clause $(\cdot)$ holds, else 0. Fig. 1 summarizes the overall embedding and detection procedure.

**Valid Bit Mechanism** To improve the robustness of our detection, we reconstruct $\boldsymbol{d}^s$ at a finer granularity by setting $l = 4$, i.e., mapped to four quantiles as opposed to two during generation. This change helps to mitigate the effects of random noise introduced into the recovered latent during recovery or under attacks, which we denote as $\hat{\boldsymbol{z}}_T^w$. We use the following quantile-based transformation to classify the latent values into $l = 4$ categories: $\boldsymbol{d}^s = \left\lfloor 4 \cdot \hat{\Phi}\left(\hat{\boldsymbol{z}}_T^w\right) \right\rfloor$, where $\hat{\Phi}(\cdot)$ is the empirical CDF of latent $\hat{\boldsymbol{z}}_T^w$. And after the inverse shuffling, we get $\boldsymbol{d}_i = \boldsymbol{d}^s_{\boldsymbol{\pi}_k^{-1}(i)}$.

To perform the bit accuracy calculation, we focus primarily on the extrema values of the distribution (i.e, low $\hat{\boldsymbol{z}}_T^w \le \hat{\Phi}(0.25)$, and high $\hat{\boldsymbol{z}}_T^w > \hat{\Phi}(0.75)$) in the sequence, as they are less likely to be altered by noise or attacks. Thus, the bit accuracy is computed as follows:

$$A_{vbit} = \frac{\sum_{i=1}^{m/2} \mathbb{I}\left(\left(\boldsymbol{d}_i = 0 \text{ and } \boldsymbol{d}_{m/2+i} = 0 \text{ or } 1\right) \text{ or } \left(\boldsymbol{d}_i = 3 \text{ and } \boldsymbol{d}_{m/2+i} = 2 \text{ or } 3\right)\right)}{\sum_{i=1}^{m/2} \mathbb{I}\left(\boldsymbol{d}_i = 0 \text{ or } \boldsymbol{d}_i = 3\right)}$$

**Expected bit accuracy under Gaussian Noise** Theorems 1 and 2 in Appendix C present the expected bit accuracy for `TabWak`, both with and without the valid bit mechanism, when the latents generated by the control sequence $\boldsymbol{d}$ are perturbed by Gaussian noise following $N(0, \sigma)$, i.e.$\hat{\boldsymbol{z}}_T^w = \boldsymbol{z}_T + \epsilon(\sigma)$, where $\epsilon \sim N(0, \sigma)$. In summary, under these conditions, the expected bit accuracy for `TabWak` without the Valid Bit Mechanism is given by

$$\mathbb{E}\left[A_{\text{bit}}\right] = \left(\int_{-\infty}^{\infty}\left[1 - \Phi\left(-\frac{|x|}{\sigma}\right)\right]\phi(x)\,dx\right)^2 + \left(\int_{-\infty}^{\infty}\Phi\left(-\frac{|x|}{\sigma}\right)\phi(x)\,dx\right)^2.$$

Similarly, the expected bit accuracy for `TabWak` with the Valid Bit Mechanism is expressed as:

$$\mathbb{E}\left[A_{\text{vbit}}\right] = 16 \left( \begin{array}{l} \left(\int_{-\infty}^{\Phi^{-1}(0.25)} \Phi\left(\frac{\Phi^{-1}(0.25)\sqrt{1+\sigma^2}+x}{\sigma}\right)\phi(x)dx\right) \times \left(\int_{-\infty}^{\Phi^{-1}(0.25)} \Phi\left(\frac{x}{\sigma}\right)\phi(x)dx\right) \\ + \left(\int_{-\infty}^{\Phi^{-1}(0.25)} \Phi\left(\frac{\Phi^{-1}(0.25)\sqrt{1+\sigma^2}-x}{\sigma}\right)\phi(x)dx\right) \times \left(\int_{-\infty}^{\Phi^{-1}(0.25)} \Phi\left(\frac{-x}{\sigma}\right)\phi(x)dx\right) \\ + \left(\int_{\Phi^{-1}(0.25)}^{0} \Phi\left(\frac{\Phi^{-1}(0.25)\sqrt{1+\sigma^2}+x}{\sigma}\right)\phi(x)dx\right) \times \left(\int_{\Phi^{-1}(0.25)}^{0} \Phi\left(\frac{x}{\sigma}\right)\phi(x)dx\right) \\ + \left(\int_{\Phi^{-1}(0.25)}^{0} \Phi\left(\frac{\Phi^{-1}(0.25)\sqrt{1+\sigma^2}-x}{\sigma}\right)\phi(x)dx\right) \times \left(\int_{\Phi^{-1}(0.25)}^{0} \Phi\left(\frac{-x}{\sigma}\right)\phi(x)dx\right) \end{array} \right).$$

Figure 2 presents the curves for $\mathbb{E}\left[A_{\text{bit}} \mid \sigma\right]$ and $\mathbb{E}\left[A_{\text{vbit}} \mid \sigma\right]$. As expected, the bit accuracy for both models is 0.5 when $\hat{\boldsymbol{z}}_T^w$ is randomly sampled from a standard Gaussian distribution (i.e., without watermarking). Moreover, it is evident that, for the same noise level $\sigma$, $\mathbb{E}\left[A_{\text{vbit}}\right]$ consistently exceeds $\mathbb{E}\left[A_{\text{bit}}\right]$. This implies that $\left(\mathbb{E}\left[A_{\text{bit}} \mid \sigma\right] - 0.5\right) \le \left(\mathbb{E}\left[A_{\text{vbit}} \mid \sigma\right] - 0.5\right)$ for $\sigma > 0$. Consequently, at equivalent noise levels, the Valid Bit Mechanism in `TabWak` introduces a greater disparity in bit accuracy compared to randomly drawn latents (non-watermarked latents). In other words, the Valid Bit Mechanism enhances robustness against noise, resulting in improved detection accuracy.

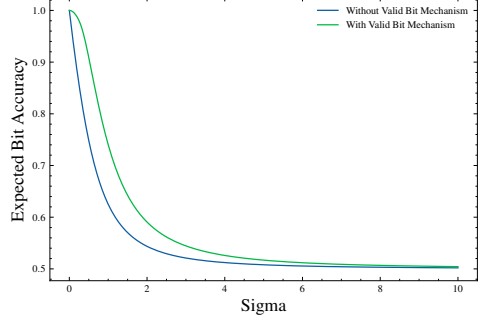

Figure 2: Comparison of expected bit accuracy with and without the Valid Bit Mechanism.

# 4 EVALUATION

## 4.1 EXPERIMENTS SETUP

**Datasets** We used five widely utilized tabular datasets to evaluate the performance of the proposed `TabWak` on synthetic data quality, its effectiveness of watermark detection, and its robustness against post-editing attacks. These include: *Shoppers* (Sakar & Kastro, 2018), *Magic* (Bock, 2007), *Credit* (Yeh, 2016), *Adult* (Becker & Kohavi, 1996), and *Diabetes* (Strack et al., 2014). Additional details regarding these datasets are provided in Appendix E.1.

**Metrics** *Data quality*: The quality of synthetic data is assessed through similarity, discriminability, and utility. Similarity measures how well the synthetic data reflects the original, focusing on shape (distribution comparisons using Kolmogorov-Smirnov and total variation distance) and trend (correlation preservation across columns). Discriminability uses logistic regression to evaluate whether a model can distinguish between synthetic and real data, with higher scores indicating better indistinguishability. Utility assesses how well synthetic data performs in machine learning tasks, using classification/regression models to compare AUC and RMSE scores. *Detectabilty*: Detectability is assessed using Z-score, which measures the difference in mean values between a synthetic table with and without watermark, and TPR@XFPR, which evaluates the True Positive Rate (TPR) at a X% False Positive Rate (XFPR) in the detection of the watermarked table.

**Tabular generative model** All experiments used a consistent latent tabular model architecture based on the Tabsyn framework Zhang et al. (2024).Given that all watermarking methods are applied during the generator's sampling phase, models for each dataset are shared across methods. Hence, for each dataset, the same generator is sampled multiple times with different watermarked latent codes to evaluate the watermark's effectiveness. Detailed model specifications are provided in Appendix B.1.

**Baselines**. To the best of our knowledge, no sampling phase watermarking technique for tabular data has yet been proposed in related work. Therefore, we adapt two commonly used watermarking techniques in image diffusion models—Tree-Ring (TR) (Wen et al., 2023) and Gaussian Shading (GS) (Yang et al., 2024)—to the tabular diffusion model. And a post-processing watermark (He et al., 2024) is also included in the Appendix F.3. Detailed implementation of these methods can be found in Appendix D.1.

## 4.2 GENERATIVE TABULAR DATA QUALITY AND WATERMARK DETECTABILITY

To evaluate the quality of the watermarked tabular data, for each comparison we generate as many rows as the original datasets. To evaluate the detectability of the watermarks, we generate a given number of rows to compute the Z-score. The mean and standard deviation across 100 tests of each quality metric and Z-score for 1K, 5K and 10K rows are presented in Table 1. These are all one-tailed Z-scores. Specifically, for Tree-Ring, the distance between watermark batches in the Fourier space for the watermarked table is expected to be smaller than for the non-watermarked table. For Gaussian Shading and our method, the bit accuracy of the watermarked table is expected to be higher than that of the non-watermarked table. The relationship between the one-tailed Z-score and p-value is illustrated in Figure 6 in the Appendix E.2.

From the results, we observe that our method consistently delivers the best or second-best quality scores both with and without the valid bit mechanism, except the Diabetes dataset where Tree-Ring at times comes in second. The quality metrics for our method are also close to those of non-watermarked data. In contrast, Gaussian Shading exhibits the worst quality scores. For instance, in all datasets, the Logistic detection score for Gaussian Shading is at least $0.4$ points worse than our proposed method. This significant drop is likely caused by Gaussian Shadings' fixed control seed across rows, resulting in less diversity in the generated tabular data.

Regarding detectability, the Gaussian Shading method shows the highest Z-scores in the Magic, Adult, Credit, and Diabetes datasets, and the second-highest Z-score in the Shoppers dataset, thanks to the shared control seed. With the valid bit mechanism, the detectability of our method improves significantly, achieving the highest Z-score in the Shoppers dataset and the second-highest in the others, except for the Adult dataset. Across all datasets and row counts, the Z-scores are consistently

Table 1: Synthetic Table Quality and Watermark Detectability: Comparison of methods without watermarking ('W/O'), Tree-Ring ('TR'), Gaussian Shading ('GS'), and `TabWak` without ('Ours') and with ('Ours*') the Valid Bit Mechanism. Best results are shown in **Bold**, and second-best results are underlined. Metrics include various quality measures and Z-scores for different row counts.

| Datasets | Method | Quality Metric | | | | Z-score | | |
|---|---|---|---|---|---|---|---|---|
| | | Shape↑ | Trend↑ | Logistic↑ | MLE↑ | 1K rows↑ | 5K rows↑ | 10K rows↑ |
| Shoppers | W/O | $0.922_{\pm0.001}$ | $0.907_{\pm0.002}$ | $0.635_{\pm0.006}$ | $0.871_{\pm0.012}$ | - | - | - |
| | TR | $0.892_{\pm0.001}$ | $0.876_{\pm0.001}$ | $0.499_{\pm0.009}$ | $0.864_{\pm0.009}$ | $3.11_{\pm0.81}$ | $6.17_{\pm0.75}$ | $6.68_{\pm0.56}$ |
| | GS | $0.767_{\pm0.001}$ | $0.716_{\pm0.001}$ | $0.166_{\pm0.006}$ | $0.816_{\pm0.031}$ | $\underline{10.02}_{\pm1.10}$ | $\underline{22.66}_{\pm0.87}$ | $\underline{31.92}_{\pm0.91}$ |
| | Ours | $\underline{0.905}_{\pm0.002}$ | $\underline{0.881}_{\pm0.001}$ | $\underline{0.523}_{\pm0.007}$ | $\mathbf{0.878}_{\pm0.010}$ | $5.42_{\pm0.88}$ | $11.89_{\pm0.95}$ | $16.94_{\pm1.09}$ |
| | Ours* | $\mathbf{0.914}_{\pm0.008}$ | $\mathbf{0.906}_{\pm0.002}$ | $\mathbf{0.580}_{\pm0.057}$ | $\underline{0.867}_{\pm0.062}$ | $\mathbf{15.46}_{\pm1.19}$ | $\mathbf{34.52}_{\pm1.05}$ | $\mathbf{48.59}_{\pm1.03}$ |
| Magic | W/O | $0.917_{\pm0.002}$ | $0.939_{\pm0.001}$ | $0.710_{\pm0.004}$ | $0.906_{\pm0.004}$ | - | - | - |
| | TR | $0.890_{\pm0.001}$ | $0.928_{\pm0.001}$ | $0.626_{\pm0.003}$ | $0.904_{\pm0.004}$ | $1.54_{\pm0.78}$ | $3.56_{\pm0.94}$ | $4.72_{\pm0.97}$ |
| | GS | $0.812_{\pm0.001}$ | $0.913_{\pm0.003}$ | $0.383_{\pm0.003}$ | $0.902_{\pm0.005}$ | $\mathbf{16.54}_{\pm0.92}$ | $\mathbf{36.77}_{\pm0.95}$ | $\mathbf{52.11}_{\pm0.89}$ |
| | Ours | $\underline{0.897}_{\pm0.002}$ | $\mathbf{0.934}_{\pm0.001}$ | $\underline{0.656}_{\pm0.007}$ | $\mathbf{0.904}_{\pm0.007}$ | $5.88_{\pm0.98}$ | $13.17_{\pm0.84}$ | $18.61_{\pm1.14}$ |
| | Ours* | $\mathbf{0.908}_{\pm0.009}$ | $0.927_{\pm0.011}$ | $\mathbf{0.705}_{\pm0.007}$ | $0.876_{\pm0.090}$ | $\underline{11.29}_{\pm1.02}$ | $\underline{25.30}_{\pm0.99}$ | $\underline{35.98}_{\pm1.08}$ |
| Adult | W/O | $0.933_{\pm0.001}$ | $0.887_{\pm0.000}$ | $0.653_{\pm0.007}$ | $0.876_{\pm0.005}$ | - | - | - |
| | TR | $0.924_{\pm0.006}$ | $0.868_{\pm0.012}$ | $0.640_{\pm0.013}$ | $0.872_{\pm0.007}$ | $0.77_{\pm0.46}$ | $0.84_{\pm0.56}$ | $0.93_{\pm0.66}$ |
| | GS | $0.732_{\pm0.001}$ | $0.487_{\pm0.024}$ | $0.023_{\pm0.001}$ | $0.858_{\pm0.004}$ | $\mathbf{18.63}_{\pm1.16}$ | $\mathbf{41.64}_{\pm1.09}$ | $\mathbf{58.60}_{\pm1.04}$ |
| | Ours | $\mathbf{0.932}_{\pm0.001}$ | $\underline{0.872}_{\pm0.001}$ | $\mathbf{0.661}_{\pm0.021}$ | $\mathbf{0.874}_{\pm0.011}$ | $\underline{15.83}_{\pm0.82}$ | $\underline{35.08}_{\pm0.93}$ | $\underline{49.70}_{\pm0.90}$ |
| | Ours* | $\underline{0.931}_{\pm0.003}$ | $\mathbf{0.884}_{\pm0.003}$ | $0.645_{\pm0.009}$ | $\mathbf{0.874}_{\pm0.008}$ | $12.58_{\pm1.09}$ | $28.45_{\pm1.06}$ | $40.15_{\pm0.95}$ |
| Credit | W/O | $0.930_{\pm0.001}$ | $0.905_{\pm0.001}$ | $0.741_{\pm0.003}$ | $0.743_{\pm0.013}$ | - | - | - |
| | TR | $0.912_{\pm0.015}$ | $0.891_{\pm0.012}$ | $0.717_{\pm0.023}$ | $0.737_{\pm0.012}$ | $2.90_{\pm0.91}$ | $5.56_{\pm1.00}$ | $6.48_{\pm0.96}$ |
| | GS | $0.566_{\pm0.001}$ | $0.655_{\pm0.001}$ | $0.129_{\pm0.002}$ | $0.715_{\pm0.016}$ | $\mathbf{37.69}_{\pm0.85}$ | $\mathbf{84.09}_{\pm1.08}$ | $\mathbf{118.81}_{\pm0.94}$ |
| | Ours | $\mathbf{0.928}_{\pm0.001}$ | $\mathbf{0.905}_{\pm0.001}$ | $\mathbf{0.750}_{\pm0.005}$ | $\underline{0.741}_{\pm0.010}$ | $4.99_{\pm1.03}$ | $11.35_{\pm0.97}$ | $15.79_{\pm1.06}$ |
| | Ours* | $\underline{0.922}_{\pm0.010}$ | $\underline{0.892}_{\pm0.016}$ | $\underline{0.677}_{\pm0.086}$ | $\mathbf{0.744}_{\pm0.009}$ | $\underline{10.10}_{\pm1.08}$ | $\underline{22.91}_{\pm1.05}$ | $\underline{32.04}_{\pm1.10}$ |
| Diabetes | W/O | $0.873_{\pm0.009}$ | $0.743_{\pm0.004}$ | $0.748_{\pm0.034}$ | $0.803_{\pm0.032}$ | - | - | - |
| | TR | $0.858_{\pm0.011}$ | $0.726_{\pm0.004}$ | $0.698_{\pm0.034}$ | $0.794_{\pm0.030}$ | $2.31_{\pm0.73}$ | $6.86_{\pm1.11}$ | $7.59_{\pm0.79}$ |
| | GS | $0.732_{\pm0.004}$ | $0.720_{\pm0.003}$ | $0.129_{\pm0.004}$ | $0.000_{\pm0.000}$ | $\mathbf{24.94}_{\pm0.80}$ | $\mathbf{55.56}_{\pm0.95}$ | $\mathbf{78.86}_{\pm0.88}$ |
| | Ours | $\mathbf{0.884}_{\pm0.007}$ | $\mathbf{0.749}_{\pm0.003}$ | $\mathbf{0.737}_{\pm0.034}$ | $0.777_{\pm0.020}$ | $\underline{2.39}_{\pm1.02}$ | $5.42_{\pm0.98}$ | $7.70_{\pm0.98}$ |
| | Ours* | $0.849_{\pm0.007}$ | $\underline{0.733}_{\pm0.004}$ | $\underline{0.694}_{\pm0.022}$ | $\mathbf{0.801}_{\pm0.039}$ | $\underline{3.95}_{\pm1.04}$ | $\underline{7.86}_{\pm0.91}$ | $\underline{11.27}_{\pm0.99}$ |

higher than $3.95$, indicating that we can detect our watermark in all cases with a false positive rate (FPR) of less than $3.9 \times 10^{-5}$.

## 4.3 ROBUSTNESS AGAINST POST-EDITING ATTACKS

For robustness against post-editing attacks, we designed five types of attacks: row deletion, column deletion, cell deletion, Gaussian noise, and shuffling. In the row deletion and cell deletion attacks, a certain percentage of rows or cells ($5\%$, $10\%$, or $20\%$) is removed. In the column deletion attack, a specific number of columns (1–3 columns) are deleted. For the Gaussian noise attack, noise is added to the numeric columns of the tabular data, where the noise's standard deviation is a percentage of the cell value. In the shuffling attack, the rows of the table are shuffled.

For the Tree-Ring watermark, which does not support row-by-row detection, we handle row deletions by replacing the deleted rows with those from a non-watermarked table. Similarly, for column and cell deletions across all watermarking methods, we replace deleted values with randomly sampled non-watermarked data to obtain the corresponding latent codes.

Table 2 presents the average Z-score for a watermarked table with 5K rows under different types of attacks across 100 tests. To reduce the impact of failed tests on the average Z-score, we set negative Z-scores in the one-tailed Z-test to zero when calculating the average. From the results, we observe that Tree-Ring, which performs *ordered* table-level detection, exhibits low robustness against row-related attacks, such as row deletion and row shuffling, with Z-scores falling below $0.6$. In contrast, Gaussian Shading and our method, which are inherently robust against these types of attacks, demonstrate superior performance. Notably, the valid bit mechanism shows strong performance. For our method without the valid bit mechanism, the Z-score drops a lot under stronger attacks. In the case of column deletion attacks, the Z-score drops to 0 in 2 out of 5 datasets. However, for our method with the valid bit mechanism, the average Z-score remains above 4 across all test conditions. In most cases, the Z-score is either the highest or second-highest among the compared methods.

Figure 3 illustrates the trade-off between detectability and data quality across various watermarking methods. The x-axis represents the theoretical false positive rate (p-value), while the y-axis shows the average of four data quality metrics (Shape, Trend, Logistic, and MLE) from Table 1, evaluated

Table 2: Robustness Against Post-Editing Attacks: Average Z-score on 5K rows, repeated 100 times, for methods without watermarking ('W/O'), Tree-Ring ('TR'), Gaussian Shading ('GS'), and `TabWak` without ('Ours') and with ('Ours*') the Valid Bit Mechanism. Best results are shown in **Bold**, and second-best results are underlined. Z-scores without attacks at 5K rows reprinted in (parentheses) from Table 1.

| Dataset | Method | | Attacks | | | | | | | | | | | | |
|---|---|---|---|---|---|---|---|---|---|---|---|---|---|---|---|
| | | | Row Deletion | | | Column Deletion | | | Cell Deletion | | | Gaussian Noise | | | Shuffling |
| | | | 5% | 10% | 20% | 1 col | 2 col | 3 col | 5% | 10% | 20% | 5% | 10% | 20% | - |
| Shoppers | TR | (6.17) | 0.37 | 0.37 | 0.36 | 3.76 | 2.59 | 1.98 | 4.87 | 4.69 | 4.27 | 3.87 | 1.48 | 0.01 | 0.00 |
| | GS | (22.66) | 21.95 | 21.36 | 20.02 | 20.99 | 19.18 | 9.19 | 22.29 | 21.96 | 22.07 | 22.25 | 21.74 | 21.64 | 22.49 |
| | Ours | (11.89) | 11.54 | 11.27 | 10.68 | 12.19 | 15.70 | 9.41 | 12.48 | 13.34 | 14.48 | 11.60 | 6.26 | 0.00 | 11.81 |
| | Ours* | (34.52) | **33.58** | **32.69** | **30.98** | **34.50** | **34.33** | **37.38** | **34.40** | **34.63** | **33.36** | **27.60** | **29.84** | **39.90** | **34.51** |
| Magic | TR | (3.56) | 0.41 | 0.34 | 0.42 | 2.29 | 1.01 | 0.42 | 4.83 | 4.53 | 4.06 | 4.95 | 4.89 | 4.38 | 0.00 |
| | GS | (36.77) | **35.97** | **35.11** | **33.07** | **35.73** | **32.80** | 34.37 | **35.37** | **34.08** | **31.96** | **36.89** | **36.91** | **36.86** | **36.92** |
| | Ours | (13.17) | 12.88 | 12.68 | 11.93 | 0.00 | 0.00 | 0.00 | 7.93 | 2.84 | 0.00 | 12.87 | 12.51 | 11.19 | 13.12 |
| | Ours* | (25.30) | 24.78 | 23.98 | 22.61 | 32.38 | 32.33 | **37.80** | 26.92 | 28.13 | 30.17 | 25.51 | 25.12 | 25.06 | 25.39 |
| Adult | TR | (0.84) | 0.39 | 0.37 | 0.36 | 0.42 | 0.43 | 0.57 | 0.30 | 0.31 | 0.37 | 0.29 | 0.67 | 1.46 | 0.00 |
| | GS | (41.64) | **40.43** | **39.40** | **37.34** | **37.94** | **48.50** | **50.79** | **43.14** | **43.55** | **43.39** | **51.02** | **66.03** | **83.84** | **41.67** |
| | Ours | (35.08) | 34.17 | 33.49 | 31.46 | 31.32 | 29.24 | 19.55 | 36.41 | 34.08 | 35.60 | 28.02 | 14.22 | 3.83 | 35.30 |
| | Ours* | (28.45) | 27.78 | 26.83 | 25.43 | 28.45 | 24.92 | 27.57 | 29.29 | 30.07 | 29.86 | 32.53 | 48.66 | 64.19 | 28.42 |
| Credit | TR | (5.56) | 0.38 | 0.39 | 0.39 | 3.31 | 1.14 | 0.57 | 5.33 | 4.81 | 3.96 | 5.70 | 5.37 | 4.75 | 0.00 |
| | GS | (84.09) | **82.20** | **79.88** | **75.45** | **84.59** | **84.49** | **88.94** | **83.78** | **84.05** | **84.21** | **74.20** | **73.16** | **76.30** | **84.36** |
| | Ours | (11.35) | 10.94 | 10.52 | 10.00 | 8.08 | 5.90 | 4.05 | 9.65 | 8.10 | 6.18 | 8.93 | 3.31 | 0.00 | 11.26 |
| | Ours* | (22.91) | 22.11 | 21.65 | 20.29 | 27.31 | 32.71 | 34.98 | 26.65 | 30.31 | 36.24 | 23.18 | 24.31 | 27.17 | 22.88 |
| Diabetes | TR | (6.86) | 0.83 | 0.42 | 0.59 | 4.60 | 3.13 | 2.15 | 6.56 | 6.27 | 5.70 | 6.51 | 5.38 | 2.37 | 0.00 |
| | GS | (55.56) | **54.21** | **52.88** | **49.73** | **55.73** | **56.94** | **58.93** | **54.09** | **52.97** | **50.72** | **53.81** | **50.59** | **49.14** | **55.94** |
| | Ours | (5.42) | 5.42 | 4.94 | 4.71 | 1.27 | 0.00 | 0.00 | 1.15 | 2.31 | 3.84 | 0.08 | 0.02 | 0.23 | 5.32 |
| | Ours* | (7.86) | 7.76 | 7.63 | 7.11 | 4.98 | 10.94 | 12.74 | 4.76 | 4.41 | 3.61 | 6.56 | 6.73 | 3.83 | 7.91 |

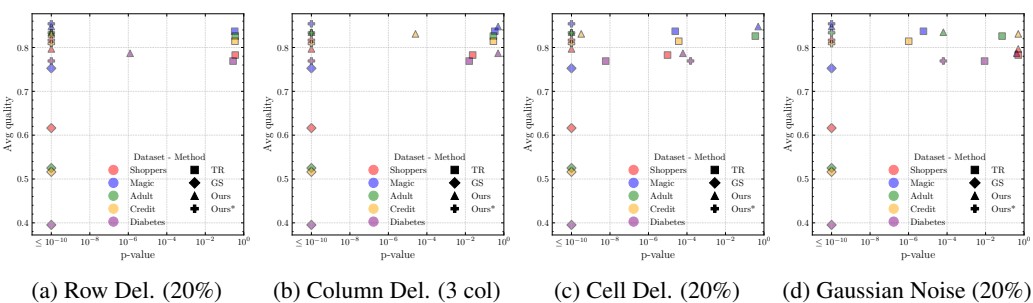

(a) Row Del. (20%)   (b) Column Del. (3 col)   (c) Cell Del. (20%)   (d) Gaussian Noise (20%)

Figure 3: The trade-off between p-value under various attacks and the average data quality

under the strongest attack settings. Notably, `TabWak` with valid bit mechanism(Ours*, indicated by filled plus markers) predominantly occupies the upper-left region, signifying superior performance in most scenarios, except under cell deletion and Gaussian noise attacks on the Diabetes dataset. In contrast, Gaussian Shading (GS), while demonstrating strong detectability, consistently appears in the lower-left region, emphasizing its compromise on data quality for robustness.

Figure 4 explores the relationship between the number of rows and TPR@0.1%FPR (True Positive Rate under 0.1% False Positive Rate). For each row count, 10 repetitive experiments were conducted on 100 tables. The strongest attacks from Table 2 were used: Cell and Noise attacks were set at 20% strength, and three columns were deleted for the Column Deletion (Del.) attack.

As shown in Figure 4, our method with the valid bit mechanism (Ours*) consistently demonstrates a significantly higher TPR compared to the version without it (Ours). Notably, our method without the valid bit mechanism underperforms, achieving a TPR@0.1%FPR below $0.5$ in 7 out of 12 cases—effectively random guessing. In contrast, our method with the valid bit mechanism achieves a $1.0$ TPR@0.1%FPR in all cases with 11 cases requiring as few as 200 rows and 1 case needing fewer than 300 rows. Moreover, our method with the valid bit mechanism shows similar or better detectability than Gaussian Shading in most scenarios while maintaining a significantly higher data quality.

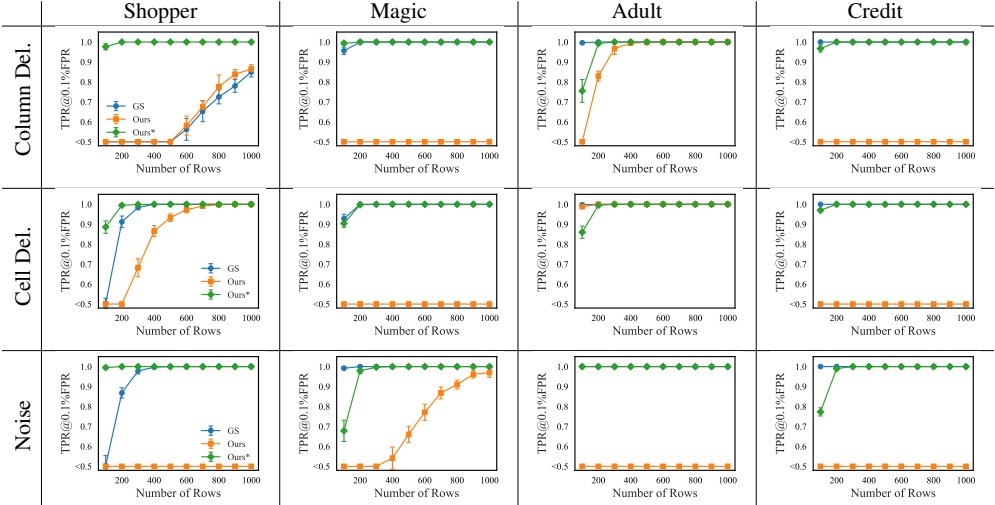

Figure 4: TPR@0.1% FPR versus row count in four datasets under various attacks. Cell and Noise attacks are set at 20% strength; Column Deletion (Del.) is fixed to three columns.

## 5 CONCLUSION

Motivated by the necessity and urgency to audit the usage of synthetic tables, we propose `TabWak`-the first row-wise watermarking scheme for tabular diffusion models. `TabWak` aims to embed an imperceptible pattern in each row, while maintaining high quality of tables and detectability in the presence of post-editing attacks. The novel feature of `TabWak` is to embed the secrete key in the positional information and values of random seeds that control Gaussian latent codes for each row, without affecting the Gaussian distribution nor limiting the sampling choices. Another feature of `TabWak` is the valid-bit detection, particularly on the tail distribution of latent embeddings. We validate the effectiveness of `TabWak` through theoretical claims and extensive experiments. Evaluation results on five datasets against image-based watermarking baselines show that `TabWak` achieves the highest tabular quality measure thanks to its diversity in latent, and resilient detectability with or without attacks.

## 6 REPRODUCIBILITY AND ETHIC STATEMENT

To ensure the reproducibility of our research, we have open-sourced the code for the various watermarking techniques and the tabular diffusion model, as shown in `https://github.com/chaoyitud/TabWak`. Furthermore, all experiments conducted as part of this study utilized publicly available datasets.

With the popularity of diffusion models and their applications, embedding watermarks into their generated content is an essential step toward trustworthy and responsible AI technology development and deployment. Our findings of improved watermark detection performance and utility provide novel insights into the research and practice of watermarking for synthetic tables.

### ACKNOWLEDGMENTS

This research was partly funded by the SNSF project, Priv-GSyn 200021E_229204, the DEPMAT project (with project number P20-22 / N21022) of the research programme Perspectief which is partly financed by the Dutch Research Council (NWO), by the Spoke "FutureHPC & BigData" of the ICSC-Centro Nazionale di Ricerca in "High Performance Computing, Big Data and Quantum Computing", funded by the European Union - NextGenerationEU, and by the DYMAN project funded by the European Union - European Innovation Council under G.A. n. 101161930. It is also part of the Partnership Program of the Materials innovation institute M2i (www.m2i.nl). We would like to express our gratitude to all involved in these projects for their support and contributions.

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

# A    NOMENCLATURE

$\boldsymbol{\pi}_\kappa$    Pseudo-random permutation applied to the control seed $\boldsymbol{d}$, indexed by the watermark key $\kappa$, used to shuffle the sequence for embedding.

$\boldsymbol{d}$    Control seed used for watermark embedding, containing integer values that define specific segments of the latent space.

$\boldsymbol{d}^s$    Shuffled control seed, generated by applying a pseudo-random permutation to $\boldsymbol{d}$, ensuring row-wise detectability of the watermark.

$\boldsymbol{z}_T$    Noisy latent variable at the final time step $T$, from which the tabular data generation starts.

$\boldsymbol{z}_t$    Latent variable at time step $t$, representing intermediate states during the diffusion process.

$\boldsymbol{z}_T^w$    Watermarked latent variable at time step $T$, altered for embedding watermark information.

$\epsilon(\sigma)$    Gaussian noise added during the perturbation process, drawn from $N(0, \sigma^2)$.

$\mathbb{I}(\cdot)$    Indicator function, used to compare whether conditions are met (returns 1 if true, 0 otherwise).

$\mathcal{D}$    Decoder, used to reconstruct tabular data from the latent space.

$\mathcal{E}$    Encoder, used to map tabular data into the latent space.

$\Phi(x)$    Cumulative distribution function (CDF) of the standard normal distribution, used to partition latent space into segments.

$\phi(x)$    Probability density function (PDF) of the standard normal distribution.

$\sigma$    Noise level in the Gaussian noise distribution, controlling the magnitude of perturbations in the watermark detection process.

$A_{\text{bit}}$    Bit accuracy, the proportion of bits correctly recovered after noise perturbation, measuring the effectiveness of the watermark embedding.

$A_{\text{vbit}}$    Valid bit accuracy, a refined accuracy measure focusing on extreme values in the bit sequence to increase resilience against noise.

$F(\boldsymbol{z}_T)$    Fourier-transformed version of the noisy latent code, used for injecting or detecting the watermark.

$K$    Watermark patch applied to the latent space, created through a structured pattern of concentric circles or ripples.

$r$    Radius in the Tree-Ring watermark, defining the size of the watermark region within the latent space.

Fourier Transform (fft2d)    A mathematical transformation used to convert the latent noise matrix into frequency space for watermark embedding.

# B    DIFFUSION AND DIFFUSION INVERSION

## B.1    TABULAR LATENT DIFFUSION MODEL

In this paper, we adopt the Tabsyn framework from (Zhang et al., 2024), which combines an autoencoding framework with a diffusion process in the latent space. This architecture efficiently manages the complexity of mixed-type tabular data, which includes both numerical and categorical variables, by encoding them into a unified latent space where the diffusion process operates. Below, we outline the key components of this architecture. Unlike Tabsyn, which utilizes a score-based diffusion process, we employ a Denoising Diffusion Probabilistic Model (DDPM) (Ho et al., 2020) for training, and leverage Denoising Diffusion Implicit Models (DDIM) (Song et al., 2021) for the sampling process.

**Autoencoding Framework** To capture the structure of tabular data, we use a Variational Autoencoder (VAE) that maps the data into latent space through tokenization, encoding, and decoding. The model handles both numerical and categorical columns by tokenizing each type: numerical features are linearly transformed into embeddings, while categorical features are one-hot encoded and embedded using a lookup table. This unified representation is then passed into a Transformer-based encoder, which captures inter-column dependencies and outputs latent embeddings $z$. These are sampled via the reparameterization trick and decoded back into reconstructed token embeddings.

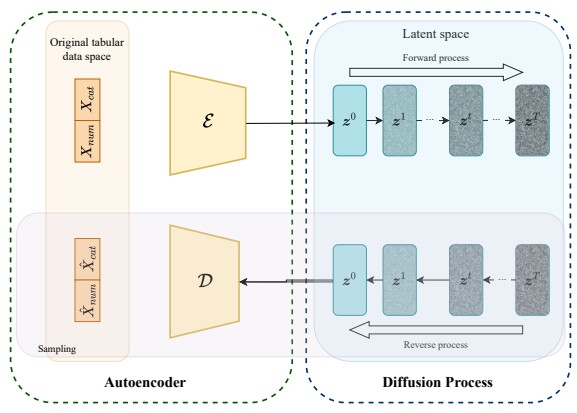

Figure 5: The diagram for tabular latent diffusion models.

Finally, the detokenizer converts these embeddings back to their original tabular form by applying inverse transformations for numerical columns and softmax for categorical columns, ensuring the output retains the original structure of the data.

**Diffusion Model** In our model, we utilize a Denoising Diffusion Probabilistic Model (DDPM) in the latent space for data generation. DDPM gradually corrupts the latent variables $z_0$ by adding Gaussian noise over a series of time steps, resulting in $z_T$, which follows a simple Gaussian distribution. During the reverse process, the model learns to denoise these latent variables step-by-step, starting from $z_T$ and progressively removing the noise to recover the original latent representation $z_0$. This reverse process is parameterized using a neural network that predicts the noise at each step, allowing the model to generate new latent variables that are then decoded back into synthetic tabular data. The DDPM approach provides high flexibility and generates diverse samples by learning to capture the complex distribution of the latent space.

More in detail for the model we used, the autoencoder module comprises an encoder and a decoder, each following a 2-layer Transformer architecture. The hidden dimension of the Transformer's feed-forward network (FFN) is set to 128. The diffusion model comprises a 4-layer multi-layer perceptron (MLP) with a hidden dimension of 1024. For both the diffusion and sampling processes within the diffusion model, 1000 timesteps are used. With these hyperparameters, the latent tabular model consistently generates high-quality synthetic data in the absence of watermarking, achieving similarity metrics above 0.88, discriminability metrics above 0.63, and utility metrics around 0.79 across all datasets. Therefore, the same architecture is employed for all four datasets, while the number of training epochs is tuned for each dataset individually.

### B.2 DDIM AND DDIM INVERSION

The diffusion model denoises a latent representation of the tabular data from a noise matrix, which can be infused with a watermark and later detected the watemarking pattern. While the architecture is oblivious to the specific choice of autoencoder architecture, the choice of diffusion model requires careful consideration to guarantee deterministic diffusion and sampling processes. Ensuring both deterministic processes allows for accurate recovery of the noise matrix from the synthesized table, thereby enabling sound detection of the watermark.

Among the various diffusion models, Denoising Diffusion Implicit Model (DDIM) (Song et al., 2021) stands out for its ability to facilitate both deterministic diffusion and sampling processes. DDIM extends the classical Markovian diffusion process into a broader class of non-Markovian diffusion processes. Within the DDIM framework, given the noise matrix $z_T$ in the latent space, and a neural network $\epsilon_\theta$ that predicts the noise $\epsilon_\theta(t, z_t)$ at each diffusion time step $t$, the generation of a

sample $z_{t-1}$ from $z_t$ during the sampling process is described by the equation:

$$z_{t-1} = \sqrt{\alpha_{t-1}}(\frac{z_t - \sqrt{1-\alpha_t}\epsilon_\theta(t, z_t)}{\sqrt{\alpha_t}}) + \sqrt{1-\alpha_{t-1}-\sigma_t^2} \cdot \epsilon_\theta(t, z_t) + \sigma_t\epsilon_t$$

where $\alpha_1, \ldots, \alpha_T$ are computed from a predefined variance schedule, $\epsilon_t \sim \mathcal{N}(0, I)$ denotes standard Gaussian noise independent of $z_t$, and the $\sigma_t$ values can be varied to yield different generative processes. Specifically, by setting $\sigma_t$ to 0 for all $t$, the sampling process becomes deterministic:

$$z_{t-1} = \sqrt{\frac{\alpha_{t-1}}{\alpha_t}}z_t + (\sqrt{1-\alpha_{t-1}} - \sqrt{\frac{\alpha_{t-1}}{\alpha_t} - \alpha_{t-1}})\epsilon_\theta(t, z_t)$$

This deterministic sampling process ensures that a given noise matrix $z_T$ consistently generates the same latent matrix $z_0$. Consequently, when $z_0$ is fed into the decoder $\mathcal{D}$, the resulting table $X = \mathcal{D}(z_0)$ will also be consistently the same.

Notably, in the limit of small steps (large value of $T$), we can traverse the timesteps in the reverse direction towards increasing levels of noise, yielding a deterministic diffusion process from $z_0$ to $z_T$, i.e. DDIM inversion:

$$z_{t+1} = \sqrt{\frac{\alpha_{t+1}}{\alpha_t}}z_t + (\sqrt{1-\alpha_{t+1}} - \sqrt{\frac{\alpha_{t+1}}{\alpha_t} - \alpha_{t+1}})\epsilon_\theta(t, z_t)$$

Therefore, given the tabular latent $z_0 = \mathcal{E}(X)$ of a table $X$, the noise matrix $z_T$ that is used to sample the corresponding table can be derived. This latent tabular diffusion model with deterministic sampling and diffusion processes enables the secure watermarking of synthetic tabular data. By embedding the watermark into the noise matrix $z_T$, the watermark remains imperceptible to humans and exerts minimal influence on the quality of the synthetic tables. By reversing the tabular data back to the noise matrix, the watermark's presence can be smoothly detected by assessing watermarking patterns.

## B.3 DECODER INVERSION

The inversion of AutoEncoder in the watermarking process is essential. However, when an inverse transformation is needed (e.g., to map a generated table back to its latent representation), simply applying the encoder ($\mathcal{E}$) to the table is insufficient due to inherent reconstruction errors. This discrepancy arises because the encoder is not the exact inverse of the decoder, meaning that using $\mathcal{E}$ for inversion leads to imperfect recovery of the latent representation. This results in a lower-bound reconstruction error defined as:

$$\|\mathcal{D}(\mathcal{E}(X)) - X\|.$$

To overcome this limitation and reduce reconstruction errors, an exact inversion of the decoder is required, ensuring that the latent representation aligns more closely with the original data. Exact inversion can enable more accurate reconstructions, enhancing performance in downstream tasks such as editing, manipulation, or generating variations of the original input.

To perform decoder inversion, we use an iterative optimization process based on gradient descent. The goal is to find the latent variable $z$ that, when passed through the decoder $\mathcal{D}$, minimizes the reconstruction error between the original table $X$ and the generated output $\mathcal{D}(z_T)$. The process starts by initializing $z$ with the output of the encoder:

$$z_T \leftarrow \mathcal{E}(X).$$

Then, we iteratively adjust $z_T$ by performing gradient descent on the objective function:

$$\|X - \mathcal{D}(z_T)\|_2^2.$$

This optimization updates the latent variable in the direction that reduces the difference between the original table and the decoder's output. The process continues until convergence, i.e., when

further updates to $z_T$ no longer significantly reduce the error. The gradient descent step can be mathematically expressed as:

$$z_T \leftarrow z_T - \eta \nabla_{z_T} \|X - \mathcal{D}(z_T)\|_2^2,$$

where $\eta$ is the learning rate. Once the process converges, the optimized latent variable $z_T$ is returned as the exact inverse representation, yielding a more accurate result than using the encoder alone.

## C  THEOREMS AND PROOFS

**Theorem 1** *Let $d \in \{0,1\}^m$ be a 1-bit string consisting of $m$ bits, where each bit follows a random Bernoulli distribution, i.e., $d_i \sim Bernoulli(1/2)$ for each $i \in \{1, \ldots, m\}$.*

*Let the perturbed Gaussian sampling process as follows:*

$$z_i = \Phi^{-1}\left(\frac{u_i + d_k}{2}\right), \quad u_i \sim \mathcal{U}(0,1),$$

*where $\Phi^{-1}$ is the inverse cumulative distribution function (CDF) of the standard normal distribution $\mathcal{N}(0,1)$, and $u_i$ is a uniform random variable.*

*Let the perturbation noise $\epsilon^1, \epsilon^2 \sim \mathcal{N}(0, \sigma^2)$ be independent Gaussian noise with variance $\sigma^2$.*

*Let the recovered bit strings as $\hat{d}_i^1$ and $\hat{d}_i^2$ after perturbation as:*

$$\hat{d}_i^j = \left\lfloor 2 \cdot F(z_T^j + \epsilon^j) \right\rfloor, \quad j = 1, 2,$$

*where $F(\cdot)$ is the empirical cumulative distribution function (CDF) of the perturbed Gaussian noise.*

*Let the **Bit Accuracy**, denoted by $A_{bit}$, as the proportion of recovered bits that match between two independent instances of the bit recovery process:*

$$A_{bit} = \frac{1}{m} \sum_{i=1}^{m} \mathbb{I}\left(\hat{d}_i^1 = \hat{d}_i^2\right),$$

*where $\mathbb{I}(\cdot)$ is the indicator function.*

*We can show that the expected value of the Bit Accuracy, $\mathbb{E}[A_{bit}]$, is given by:*

$$\mathbb{E}[A_{bit}] = \left(\int_{-\infty}^{\infty} \left[1 - \Phi\left(-\frac{|x|}{\sigma}\right)\right] \phi(x)\, dx\right)^2 + \left(\int_{-\infty}^{\infty} \Phi\left(-\frac{|x|}{\sigma}\right) \phi(x)\, dx\right)^2,$$

*where $\Phi(x)$, $\phi(x)$ is the CDF and PDF of the standard normal distribution.*

**Proof:**  When the string $d$ consists of a single bit, the bit string recovery process simplifies. Specifically, the recovered bit $\hat{d}_i$ can be written as:

$$\hat{d}_i = \begin{cases} 1 & \text{if } z_i + \epsilon_i \geq 0, \\ 0 & \text{if } z_i + \epsilon_i < 0, \end{cases}$$

where $z_i$ follows the perturbed Gaussian process and $\epsilon_i \sim \mathcal{N}(0, \sigma^2)$ is Gaussian noise.

Thus, the expected Bit Accuracy can be interpreted as the probability that the signs of $z_i^1 + \epsilon_i^1$ and $z_i^2 + \epsilon_i^2$ agree, where $z_i^1$ and $z_i^2$ are two independent instances of the perturbed Gaussian process. Mathematically, this is expressed as:

$$\mathbb{E}\left[A_{\text{bit}}\right] = \mathbb{E}\left[\mathbb{P}\left(\text{sign}(z_i^1 + \epsilon_i^1) = \text{sign}(z_i^2 + \epsilon_i^2)\right)\right].$$

The probability that the signs match can be decomposed into two cases:

1) The signs of both $z_i^1 + \epsilon_i^1$ and $z_i^2 + \epsilon_i^2$ are positive. 2) The signs of both $z_i^1 + \epsilon_i^1$ and $z_i^2 + \epsilon_i^2$ are negative.

Thus, we have:

$$\mathbb{E}\left[A_{\text{bit}}\right] = \mathbb{E}[\mathbb{P}\left(\text{no flip for } \boldsymbol{z}_i^1\right) \cdot \mathbb{P}\left(\text{no flip for } \boldsymbol{z}_i^2\right) + \mathbb{P}\left(\text{flip for } \boldsymbol{z}_i^1\right) \cdot \mathbb{P}\left(\text{flip for } \boldsymbol{z}_i^2\right)]$$

Using the Gaussian CDF $\Phi(x)$, this becomes:

$$\mathbb{E}\left[A_{\text{bit}}\right] = \int_{-\infty}^{\infty} \int_{-\infty}^{\infty} \left[\left(1 - \Phi\left(-\frac{|x_1|}{\sigma}\right)\right)\left(1 - \Phi\left(-\frac{|x_2|}{\sigma}\right)\right) + \Phi\left(-\frac{|x_1|}{\sigma}\right)\Phi\left(-\frac{|x_2|}{\sigma}\right)\right] \phi(x_1)\phi(x_2)\, dx_1\, dx_2,$$

where:

- $\Phi(\cdot)$ is the CDF of the standard normal distribution.
- $\phi(x) = \frac{1}{\sqrt{2\pi}}e^{-x^2/2}$ is the probability density function (PDF) of the standard normal distribution.

This simplifies to:

$$\mathbb{E}\left[A_{\text{bit}}\right] = \left(\int_{-\infty}^{\infty}\left[1 - \Phi\left(-\frac{|x|}{\sigma}\right)\right]\phi(x)\, dx\right)^2 + \left(\int_{-\infty}^{\infty}\Phi\left(-\frac{|x|}{\sigma}\right)\phi(x)\, dx\right)^2.$$

**Theorem 2** *Let $\boldsymbol{d} \in \{0,1,2,3\}^m$ represent an $m$-length string, where each element $\boldsymbol{d}_i$ is an independent random variable following a categorical distribution over the set $\{0,1,2,3\}$. Specifically, for each $i \in \{1,\ldots,m\}$, the random variable $\boldsymbol{d}_i$ is distributed according to Categorical$(p_0,p_1,p_2,p_3)$, with $p_0 = p_1 = p_2 = p_3 = \frac{1}{4}$.*

*Define the perturbed Gaussian sampling process as follows:*

$$\boldsymbol{z}_i = \Phi^{-1}\left(\frac{u_i + \boldsymbol{d}_k}{4}\right), \quad u_i \sim \mathcal{U}(0,1),$$

*where $\Phi^{-1}$ is the inverse cumulative distribution function (CDF) of the standard normal distribution $\mathcal{N}(0,1)$, and $u_i$ is a uniform random variable.*

*Let the perturbation noise $\epsilon^1, \epsilon^2 \sim \mathcal{N}(0,\sigma^2)$ be independent Gaussian noise with variance $\sigma^2$.*

*Define the recovered bit strings $\hat{\boldsymbol{d}}_i^1$ and $\hat{\boldsymbol{d}}_i^2$ after perturbation as:*

$$\hat{\boldsymbol{d}}_i^j = \left\lfloor 4 \cdot F(\boldsymbol{z}_T^j + \epsilon^j)\right\rceil, \quad j = 1,2,$$

*where $F(\cdot)$ is the empirical cumulative distribution function (CDF) of the perturbed Gaussian noise.*

*We define the **Valid Bit Accuracy**, denoted $A_{bit}$, as the proportion of recovered bits that match between two independent instances of the bit recovery process:*

$$A_{vbit} = \frac{\sum_{i=1}^{m} \mathbb{I}\left(\left(\hat{\boldsymbol{d}}_i^1 = 0 \text{ and } \hat{\boldsymbol{d}}_i^2 = 0 \text{ or } 1\right) \text{ or } \left(\hat{\boldsymbol{d}}_i^1 = 3 \text{ and } \hat{\boldsymbol{d}}_i^2 = 2 \text{ or } 3\right)\right)}{\sum_{i=1}^{m} \mathbb{I}\left(\hat{\boldsymbol{d}}_i^1 = 0 \text{ or } \hat{\boldsymbol{d}}_i^1 = 3\right)}$$

*where $\mathbb{I}(\cdot)$ is the indicator function.*

*We can show that the expected value of the Bit Accuracy, $\mathbb{E}[A_{vbit}]$, is given by*

$$16 \left( \begin{array}{l} \left(\int_{-\infty}^{\Phi^{-1}(0.25)} \Phi\left(\frac{\Phi^{-1}(0.25)\sqrt{1+\sigma^2}+x}{\sigma}\right)\phi(x)\, dx\right) \times \left(\int_{-\infty}^{\Phi^{-1}(0.25)} \Phi\left(\frac{x}{\sigma}\right)\phi(x)\, dx\right) \\[2ex] + \left(\int_{-\infty}^{\Phi^{-1}(0.25)} \Phi\left(\frac{\Phi^{-1}(0.25)\sqrt{1+\sigma^2}-x}{\sigma}\right)\phi(x)\, dx\right) \times \left(\int_{-\infty}^{\Phi^{-1}(0.25)} \Phi\left(\frac{-x}{\sigma}\right)\phi(x)\, dx\right) \\[2ex] + \left(\int_{\Phi^{-1}(0.25)}^{0} \Phi\left(\frac{\Phi^{-1}(0.25)\sqrt{1+\sigma^2}+x}{\sigma}\right)\phi(x)\, dx\right) \times \left(\int_{\Phi^{-1}(0.25)}^{0} \Phi\left(\frac{x}{\sigma}\right)\phi(x)\, dx\right) \\[2ex] + \left(\int_{\Phi^{-1}(0.25)}^{0} \Phi\left(\frac{\Phi^{-1}(0.25)\sqrt{1+\sigma^2}-x}{\sigma}\right)\phi(x)\, dx\right) \times \left(\int_{\Phi^{-1}(0.25)}^{0} \Phi\left(\frac{-x}{\sigma}\right)\phi(x)\, dx\right) \end{array} \right)$$

$\Phi(\cdot)$ *is the CDF of the standard normal distribution.*

**Proof:**

$$\mathbb{E}\left[A_{\text{vbit}}\right] = \sum_{k=0}^{3} P(\boldsymbol{d}_i = k) \cdot \mathbb{E}[A \mid \boldsymbol{d}(i) = \boldsymbol{d}]$$

There are eight situations that satisfy the condition of Valid bit accuracy:

1. $\boldsymbol{d}_i = 0, \hat{\boldsymbol{d}}_i^1 = 0, \hat{\boldsymbol{d}}_i^2 = 0$ or $1$
2. $\boldsymbol{d}_i = 1, \hat{\boldsymbol{d}}_i^1 = 0, \hat{\boldsymbol{d}}_i^2 = 0$ or $1$
3. $\boldsymbol{d}_i = 2, \hat{\boldsymbol{d}}_i^1 = 0, \hat{\boldsymbol{d}}_i^2 = 0$ or $1$
4. $\boldsymbol{d}_i = 3, \hat{\boldsymbol{d}}_i^1 = 0, \hat{\boldsymbol{d}}_i^2 = 0$ or $1$
5. $\boldsymbol{d}_i = 0, \hat{\boldsymbol{d}}_i^1 = 3, \hat{\boldsymbol{d}}_i^2 = 2$ or $3$
6. $\boldsymbol{d}_i = 1, \hat{\boldsymbol{d}}_i^1 = 3, \hat{\boldsymbol{d}}_i^2 = 2$ or $3$
7. $\boldsymbol{d}_i = 2, \hat{\boldsymbol{d}}_i^1 = 3, \hat{\boldsymbol{d}}_i^2 = 2$ or $3$
8. $\boldsymbol{d}_i = 3, \hat{\boldsymbol{d}}_i^1 = 3, \hat{\boldsymbol{d}}_i^2 = 2$ or $3$

Based on the symmetric property of the Gaussian distribution, we can easily get that:

- $P(\boldsymbol{d}_i = 0, \hat{\boldsymbol{d}}_i^1 = 0, \hat{\boldsymbol{d}}_i^2 = 0$ or $1) = P(\boldsymbol{d}_i = 3, \hat{\boldsymbol{d}}_i^1 = 3, \hat{\boldsymbol{d}}_i^2 = 2$ or $3)$
- $P(\boldsymbol{d}_i = 1, \hat{\boldsymbol{d}}_i^1 = 0, \hat{\boldsymbol{d}}_i^2 = 0$ or $1) = P(\boldsymbol{d}_i = 2, \hat{\boldsymbol{d}}_i^1 = 3, \hat{\boldsymbol{d}}_i^2 = 2$ or $3)$
- $P(\boldsymbol{d}_i = 2, \hat{\boldsymbol{d}}_i^1 = 0, \hat{\boldsymbol{d}}_i^2 = 0$ or $1) = P(\boldsymbol{d}_i = 1, \hat{\boldsymbol{d}}_i^1 = 3, \hat{\boldsymbol{d}}_i^2 = 2$ or $3)$
- $P(\boldsymbol{d}_i = 3, \hat{\boldsymbol{d}}_i^1 = 0, \hat{\boldsymbol{d}}_i^2 = 0$ or $1) = P(\boldsymbol{d}_i = 0, \hat{\boldsymbol{d}}_i^1 = 3, \hat{\boldsymbol{d}}_i^2 = 2$ or $3)$

So, we split the problem into 4 situations:

1) $P(\hat{\boldsymbol{d}}_i^1 = 0$ and $(\hat{\boldsymbol{d}}_i^2 = 0$ or $\hat{\boldsymbol{d}}_i^2 = 1) \mid \boldsymbol{d}_i = 0)$

Given that $\boldsymbol{d}_i = 0$, both $\boldsymbol{z}_T^1(i)$ and $\boldsymbol{z}_T^2(i)$ are initially in the $0\% - 25\%$ quantile of the standard normal distribution. That is:

$$\boldsymbol{z}_T^1(i), \boldsymbol{z}_T^2(i) \in \left[\Phi^{-1}(0), \Phi^{-1}(0.25)\right]$$

where $\Phi^{-1}(q)$ is the inverse cumulative distribution function (CDF) of the standard normal distribution, corresponding to the $q$-quantile.

After adding independent Gaussian noise $\epsilon_i^1 \sim N(0, \sigma^2)$ and $\epsilon_i^2 \sim N(0, \sigma^2)$, the noisy signals $\boldsymbol{z}_T^1(i) + \epsilon_i^1$ and $\boldsymbol{z}_T^2(i) + \epsilon_i^2$ are distributed as $N(\boldsymbol{z}_T^1(i), \sigma^2)$ and $N(\boldsymbol{z}_T^2(i), \sigma^2)$, respectively. Thus, the combined distribution of each signal is:

$$\boldsymbol{z}_T^1(i) + \epsilon_i^1, \boldsymbol{z}_T^2(i) + \epsilon_i^2 \sim N(0, 1 + \sigma^2)$$

The condition $\hat{\boldsymbol{d}}_i^1 = 0$ implies that the noisy signal $\boldsymbol{z}_T^1(i) + \epsilon_i^1$ falls into the $0\% - 25\%$ quantile of the $N(0, 1 + \sigma^2)$ distribution. Hence, we need to compute:

$$P(\hat{\boldsymbol{d}}_i^1 = 0 \mid \boldsymbol{d}_i = 0) = P\left(\boldsymbol{z}_T^1(i) + \epsilon_i^1 < \Phi^{-1}(0.25) \cdot \sqrt{1 + \sigma^2} \mid \boldsymbol{z}_T^1(i) \in \left[\Phi^{-1}(0), \Phi^{-1}(0.25)\right]\right)$$

For any specific value $\boldsymbol{z}_T^1(i) = x$, after adding noise, the noisy signal $\boldsymbol{z}_T^1(i) + \epsilon_i^1$ is distributed as $N(x, \sigma^2)$. The conditional probability is:

$$P(\boldsymbol{z}_T^1(i) + \epsilon_i^1 < \Phi^{-1}(0.25) \cdot \sqrt{1 + \sigma^2} \mid \boldsymbol{z}_T^1(i) = x) = \Phi\left(\frac{\Phi^{-1}(0.25) \cdot \sqrt{1 + \sigma^2} - x}{\sigma}\right)$$

Therefore, the total probability $P(\hat{\boldsymbol{d}}_i^1 = 0 \mid \boldsymbol{d}_i = 0)$ is the integral over the $0\% - 25\%$ quantile:

$$P(\hat{\boldsymbol{d}}_i^1 = 0 \mid \boldsymbol{d}_i = 0) = \int_{\Phi^{-1}(0)}^{\Phi^{-1}(0.25)} \Phi\left(\frac{\Phi^{-1}(0.25) \cdot \sqrt{1 + \sigma^2} - x}{\sigma}\right) f_{\boldsymbol{z}_T^1(i)}(x)dx$$

$$= 4 \int_{-\infty}^{\Phi^{-1}(0.25)} \Phi\left(\frac{\Phi^{-1}(0.25)\sqrt{1 + \sigma^2} - x}{\sigma}\right) \phi(x)dx$$

where $f_{\boldsymbol{z}_T^1(i)}(x)$ is the probability density function (PDF) of $\boldsymbol{z}_T^1(i)$ in the $0\% - 25\%$ quantile:

$$f_{\boldsymbol{z}_T^1(i)}(x) = \frac{\phi(x)}{\int_{\Phi^{-1}(0)}^{\Phi^{-1}(0.25)} \phi(t)dt} = 4\phi(x)$$

with $\phi(x) = \frac{1}{\sqrt{2\pi}}e^{-x^2/2}$ being the standard normal PDF.

The condition $\hat{\boldsymbol{d}}_i^2 = 0$ or $\hat{\boldsymbol{d}}_i^2 = 1$ means that the noisy signal $\boldsymbol{z}_T^2(i) + \epsilon_i^2$ falls into either the $0\% - 25\%$ or the $25\% - 50\%$ quantile of the new distribution $N(0, 1 + \sigma^2)$. The respective probabilities are:

- For $\hat{\boldsymbol{d}}_i^2 = 0$ (0%-25% quantile):

$$P(\hat{\boldsymbol{d}}_i^2 = 0 \mid \boldsymbol{z}_T^2(i) = x) = \Phi\left(\frac{\Phi^{-1}(0.25) \cdot \sqrt{1+\sigma^2} - x}{\sigma}\right)$$

- For $\hat{\boldsymbol{d}}_i^2 = 1$ (25%-50% quantile):

$$P(\hat{\boldsymbol{d}}_i^2 = 1 \mid \boldsymbol{z}_T^2(i) = x) = \Phi\left(\frac{\Phi^{-1}(0.50) \cdot \sqrt{1+\sigma^2} - x}{\sigma}\right) - \Phi\left(\frac{\Phi^{-1}(0.25) \cdot \sqrt{1+\sigma^2} - x}{\sigma}\right)$$

Thus, the total probability $P(\hat{\boldsymbol{d}}_i^2 = 0$ or $\hat{\boldsymbol{d}}_i^2 = 1 \mid \boldsymbol{d}_i = 0)$ is:

$$P\left(\hat{\boldsymbol{d}}_i^2 = 0 \text{ or } \hat{\boldsymbol{d}}_i^2 = 1 \mid \boldsymbol{d}_i = 0\right) = \int_{\Phi^{-1}(0)}^{\Phi^{-1}(0.25)} \left(\Phi\left(\frac{\Phi^{-1}(0.25) \cdot \sqrt{1+\sigma^2} - x}{\sigma}\right)\right.$$
$$+ \left.\left(\Phi\left(\frac{\Phi^{-1}(0.50) \cdot \sqrt{1+\sigma^2} - x}{\sigma}\right) - \Phi\left(\frac{\Phi^{-1}(0.25) \cdot \sqrt{1+\sigma^2} - x}{\sigma}\right)\right)\right) f_{\boldsymbol{z}_T^2(i)}(x)dx$$
$$= 4\left(\int_{-\infty}^{\Phi^{-1}(0.25)} \Phi\left(\frac{-x}{\sigma}\right)\phi(x)dx\right)$$

The total probability $P(\hat{\boldsymbol{d}}_i^1 = 0$ and $(\hat{\boldsymbol{d}}_i^2 = 0$ or $\hat{\boldsymbol{d}}_i^2 = 1) \mid \boldsymbol{d}_i = 0)$ is the product of the two integrals:

$$P(\hat{\boldsymbol{d}}_i^1 = 0 \text{ and } (\hat{\boldsymbol{d}}_i^2 = 0 \text{ or } \hat{\boldsymbol{d}}_i^2 = 1) \mid \boldsymbol{d}_i = 0) =$$
$$16\left(\int_{-\infty}^{\Phi^{-1}(0.25)} \Phi\left(\frac{\Phi^{-1}(0.25)\sqrt{1+\sigma^2} - x}{\sigma}\right)\phi(x)dx\right)\left(\int_{-\infty}^{\Phi^{-1}(0.25)} \Phi\left(\frac{-x}{\sigma}\right)\phi(x)dx\right)$$

Similarly, we can get

$$P\left(\hat{\boldsymbol{d}}_i^1 = 0 \text{ and } \left(\hat{\boldsymbol{d}}_i^2 = 0 \text{ or } \hat{\boldsymbol{d}}_i^2 = 1\right) \mid \boldsymbol{d}_i = 1\right) =$$
$$16\left(\int_{\Phi^{-1}(0.25)}^{0} \Phi\left(\frac{\Phi^{-1}(0.25)\sqrt{1+\sigma^2} + x}{\sigma}\right)\phi(x)dx\right)\left(\int_{\Phi^{-1}(0.25)}^{0} \Phi\left(\frac{x}{\sigma}\right)\phi(x)dx\right)$$

$$P\left(\hat{\boldsymbol{d}}_i^1 = 0 \text{ and } \left(\hat{\boldsymbol{d}}_i^2 = 0 \text{ or } \hat{\boldsymbol{d}}_i^2 = 1\right) \mid \boldsymbol{d}_i = 2\right) =$$
$$16\left(\int_{-\infty}^{\Phi^{-1}(0.25)} \Phi\left(\frac{\Phi^{-1}(0.25)\sqrt{1+\sigma^2} - x}{\sigma}\right)\phi(x)dx\right)\left(\int_{-\infty}^{\Phi^{-1}(0.25)} \Phi\left(\frac{-x}{\sigma}\right)\phi(x)dx\right)$$

$$P\left(\hat{\boldsymbol{d}}_i^1 = 0 \text{ and } \left(\hat{\boldsymbol{d}}_i^2 = 0 \text{ or } \hat{\boldsymbol{d}}_i^2 = 1\right) \mid \boldsymbol{d}_i = 3\right) =$$
$$16\left(\int_{-\infty}^{\Phi^{-1}(0.25)} \Phi\left(\frac{\Phi^{-1}(0.25)\sqrt{1+\sigma^2} + x}{\sigma}\right)\phi(x)dx\right)\left(\int_{-\infty}^{\Phi^{-1}(0.25)} \Phi\left(\frac{x}{\sigma}\right)\phi(x)dx\right)$$

The final $\mathbb{E}\left[A_{\text{vbit}}\right]$ can be summarized as:

$$\mathbb{E}\left[A_{\text{vbit}}\right] =$$

$$2\left(P\left(\hat{\boldsymbol{d}}_i^1 = 0 \text{ and } (\hat{\boldsymbol{d}}_i^2 = 0 \text{ or } \hat{\boldsymbol{d}}_i^2 = 1) \mid \boldsymbol{d}_i = 0\right) + P\left(\hat{\boldsymbol{d}}_i^1 = 0 \text{ and } (\hat{\boldsymbol{d}}_i^2 = 0 \text{ or } \hat{\boldsymbol{d}}_i^2 = 1) \mid \boldsymbol{d}_i = 1\right)\right.$$

$$\left. + P\left(\hat{\boldsymbol{d}}_i^1 = 0 \text{ and } (\hat{\boldsymbol{d}}_i^2 = 0 \text{ or } \hat{\boldsymbol{d}}_i^2 = 1) \mid \boldsymbol{d}_i = 2\right) + P\left(\hat{\boldsymbol{d}}_i^1 = 0 \text{ and } (\hat{\boldsymbol{d}}_i^2 = 0 \text{ or } \hat{\boldsymbol{d}}_i^2 = 1) \mid \boldsymbol{d}_i = 3\right)\right)/2$$

$$= 16 \begin{pmatrix} \left(\int_{-\infty}^{\Phi^{-1}(0.25)} \Phi\left(\frac{\Phi^{-1}(0.25)\sqrt{1+\sigma^2}+x}{\sigma}\right)\phi(x)dx\right) \times \left(\int_{-\infty}^{\Phi^{-1}(0.25)} \Phi\left(\frac{x}{\sigma}\right)\phi(x)dx\right) \\ + \left(\int_{-\infty}^{\Phi^{-1}(0.25)} \Phi\left(\frac{\Phi^{-1}(0.25)\sqrt{1+\sigma^2}-x}{\sigma}\right)\phi(x)dx\right) \times \left(\int_{-\infty}^{\Phi^{-1}(0.25)} \Phi\left(\frac{-x}{\sigma}\right)\phi(x)dx\right) \\ + \left(\int_{\Phi^{-1}(0.25)}^{0} \Phi\left(\frac{\Phi^{-1}(0.25)\sqrt{1+\sigma^2}+x}{\sigma}\right)\phi(x)dx\right) \times \left(\int_{\Phi^{-1}(0.25)}^{0} \Phi\left(\frac{x}{\sigma}\right)\phi(x)dx\right) \\ + \left(\int_{\Phi^{-1}(0.25)}^{0} \Phi\left(\frac{\Phi^{-1}(0.25)\sqrt{1+\sigma^2}-x}{\sigma}\right)\phi(x)dx\right) \times \left(\int_{\Phi^{-1}(0.25)}^{0} \Phi\left(\frac{-x}{\sigma}\right)\phi(x)dx\right) \end{pmatrix}$$

## D IMPLEMENTATION DETAILS

### D.1 TREE-RING

The Tree-Ring watermark is specifically designed for images, taking into account the unique characteristics of image data. In image synthesis, images and it's latent code are typically square, such as a $256 \times 256$ matrix. The Tree-Ring watermark is centrally placed within the image and resembles concentric rings, much like the rings of a tree.

In contrast, tabular data usually consists of far more rows than columns, resulting in a tall rectangular shape, such as a $10000 \times 40$ matrix. We use a different shape for Tree-Ring watermark in tabular data, whose shape is more like a ripple. For a predefined radius $r$ representing the outermost circle of the ripple watermark, a watermark patch $K$ is generated with the ripple originating from its center using Algorithm 1. The process starts by generating a random matrix of the same size as the noisy latent code using Gaussian noise. This matrix undergoes a 2D Fourier transform, with the zero-frequency component shifted to the center, providing the base for the watermark patch. The ripple is then created, where each concentric circle shares the same value, sampled randomly from the transformed base matrix. The radius $r$ is determined as 10% of the number of rows of the table in a generation.

---

**Algorithm 1** Tree-Ring Embedding in Tabular Data

---

1: **Input:** Radius $r$, shape $(n, m)$ of the noise matrix where $n$ is the number of rows and $m$ is the dimension of the latent for each row
2: $N \leftarrow (N_{ij} \sim \mathcal{N}(0,1))_{1 \leq i \leq n, 1 \leq j \leq m}$         // Initialize a random Gaussian matrix
3: $K \leftarrow \text{fftshift}(\text{fft2d}(N))$     // Apply 2D Fourier transform to $N$ and shift the components
4: $K_{\text{tmp}} \leftarrow \text{copy}(K)$         // Copy $K$ for value sampling in ripple circles
5: **for** $k \leftarrow r$ **downto** $1$ **do**
6:                                     // From the outermost circle inward
7:     $v \leftarrow \text{sample}(K_{\text{tmp}})$         // Sample a random value $v$ for the $i$-th circle
8:     **for all** $(i, j)$ where $1 \leq i \leq n, 1 \leq j \leq m, (i - \frac{n}{2})^2 + (j - \frac{m}{2})^2 \leq k^2$ **do**
9:         $K(i, j) \leftarrow v$         // Assign $v$ to position $(i, j)$
10:     **end for**
11: **end for**
12: **return** $K$         // Return the modified matrix

---

The watermarking injection and detection phases proceed as follows. In the injection phase, specific matrix elements in the Fourier-transformed noisy latent code are replaced by values from a predefined watermark patch. Specifically, given the predefined watermark patch $K$ and a binary mask $M$ with values of 1 in the ripple watermark region and 0 elsewhere, the FFT-transformed noise matrix $F(\boldsymbol{z}_T)$ of the initial noisy latent code $\boldsymbol{z}_T$ is watermarked as follows:

$$F(\boldsymbol{z}_T)_{i,j} = \begin{cases} K_{i,j}, & \text{if } M_{i,j} = 1 \\ F(\boldsymbol{z}_T)_{i,j}, & \text{otherwise} \end{cases} \quad \text{where } M_{i,j} = \begin{cases} 1, & \text{if } (i - \frac{n}{2})^2 + (j - \frac{m}{2})^2 \leq r^2 \\ 0, & \text{otherwise} \end{cases}$$

The watermarked FFT-transformed noisy latent code is then subjected to an inverse fast Fourier transform (IFFT). This process generates a synthetic table with the watermark embedded, produced through the sampling and decoding processes of the diffusion model.

In the detection phase, the synthetic tables are encoded back into the latent space and diffused to the noisy latent code $\tilde{z}_T$. An FFT is applied to this code, and the detection process compares the ground-truth watermark patch $K$ with the watermarked region of the Fourier-transformed latent noise matrix $F(\tilde{z}_T)$ using the $L_1$ distance. The distance metric is defined as:

$$\text{Dist} = \frac{1}{|M|} \sum_{M_{i,j}=1} |K_{i,j} - F(\tilde{z}_T)_{i,j}|$$

### D.2 Gaussian Shading

To adapt Gaussian Shading from the image domain to the tabular data domain, unlike Tree-Ring, which embeds a single watermark across different rows, we embed the watermark on a per-row basis. This approach benefits row-by-row detection and enhances robustness against row-wise attacks.

However, we are constrained to using the same control seed $d$ for all rows. Using multiple control seeds would require additional information, such as row indices, to generate different control seeds for each row. This method, however, would not be robust against row-wise attacks, as information like row indices can easily be altered by operations such as row deletion or shuffling.

## E Experiments Details

### E.1 Datasets

The overview of the datasets use are shown in Table 3. The **Shoppers** (Sakar & Kastro, 2018) captures online shoppers' purchasing intentions via 12,330 samples featuring 18 mixed-type columns (10 continuous and 8 categorical).The **Magic** (Bock, 2007) dataset simulates the registration of high-energy gamma particles and consists of 19,020 instances with 11 columns (10 continuous and 1 categorical). The **Credit** (Yeh, 2016) dataset provides data on the default payments of credit card clients, comprising 30,000 instances with a total of 24 mixed-type columns (14 continuous and 10 categorical). The **Adult** (Becker & Kohavi, 1996) dataset contains information on individuals' annual incomes, consisting of 48,842 instances with 15 mixed-type columns (6 continuous and 9 categorical). Finally, the **Diabetes** (Strack et al., 2014) dataset contains medical information on Pima Indian female patients aged 21 or older, consisting of 768 instances with 9 columns (8 continuous and 1 categorical).

| Name | Domain | # Rows | # Cat | # Num | Task | Target Column |
|---|---|---|---|---|---|---|
| Shoppers | Retail | 12,330 | 8 | 10 | Classification | "Revenue" |
| Magic | Physics | 19,019 | 1 | 10 | Classification | "class" |
| Adult | Social Science | 48,842 | 9 | 6 | Classification | "class" |
| Credit | Finance | 30,000 | 11 | 14 | Classification | "default" |
| Diabetes | Medical | 768 | 1 | 8 | Classification | "Outcome" |

Table 3: Properties of selected datasets used in the evaluation. # Rows, # Cat, # Num indicate the number of rows, the number of categorical columns, and the number of numerical columns, respectively. # Test indicates the number of samples in the test set.

### E.2 Metric

The quality of the synthetic data is evaluated on three aspects: similarity, discriminability, and utility.

**Similarity** This aspect assesses the statistical similarity between the synthetic data and the original data through the following metrics:

- **Shape quality**: This metric evaluates resemblance by comparing the distribution of each column in the synthetic data with its counterpart in the original data. A similarity score is computed for each column based on the distributions in both datasets. In particular, we use the complement of the Kolmogorov-Smirnov statistic for continuous columns and of the total variation distance for categorical columns, respectively (Dat, 2023). The average of these scores across all columns reflects the shape quality of the synthetic table. A higher score indicates a greater similarity between the synthetic and original tables.

- **Trend quality**: Since columns in the data may or may not relate to each other, this metric assesses how well the synthetic data preserves these relationships. This involves calculating the correlation between every pair of columns in both the synthetic and original datasets and comparing them. A higher trend quality score indicates that the synthetic data accurately represents the inter-column relationships found in the original data. The average of these scores across all column pairs is used to represent the trend quality of the synthetic table.

**Discriminability**    This aspect assesses how difficult it is for a machine learning model to distinguish between synthetic data and original data through the following method:

- **Logistic Detection**: A logistic regression model is trained to differentiate between the two datasets. First, all rows from both the real and synthetic datasets are combined and then split into training and validation sets. The machine learning model is trained on the training set and evaluated on the validation sets. The performance of the model is measured based on the averaged complement of ROC AUC score across all validation splits. A higher score indicates that the model can not differentiate between synthetic and real data, suggesting higher in-discriminability for the synthetic data, i.e. synthetic data being undistinguishable from real data.

**Utility**    This aspect evaluates the quality of the synthetic data in terms of their performance in downstream machine-learning tasks. This evaluation is conducted using the following method:

- **Machine Learning Efficacy (MLE)**: Following the training-on-synthetic and test-on-real setting, each dataset's classification or regression model is trained on the synthetic data and evaluated using the real testing set. The performance of the model is measured by the AUC score for classification tasks and the RMSE for regression tasks. A higher MLE score indicates better machine learning utility of the synthetic data.

**Detectability**    Detectability assesses how well watermarks can be detected in synthetic data. We use two metrics: Z-score and TPR@XFPR.

- **Z-score**: This metric measures the difference in a watermarking statistic between watermarked and non-watermarked tables. For row-wise watermarks (e.g., Gaussian Shading and `TabWak`), the metric is the bit accuracy for each row. For Tree-Ring, we calculate the distance between the watermark patch in the Fourier domain. We compute the empirical mean and variance of these metrics for non-watermarked tables as a baseline. For row-wise watermarking, such as Gaussian Shading and `TabWak`, we use 500,000 rows to calculate statistics of non-watermarked rows, assuming the bit accuracy for watermarked rows is higher. For Tree-Ring watermarking, the statistics of non-watermarked tables is computed over 1,000 tables per shape, based on the assumption that watermarked tables have a smaller distance between watermark patches than non-watermarked ones. In both cases, the Z-score is used for a one-tailed test to reject the null hypothesis that the table is non-watermarked. The Z-score for a table with $n$ rows for row-wise watermarking is calculated as: The Z-score measures the difference between the mean bit accuracy of watermarked rows and non-watermarked rows, normalized by the standard deviation of the non-watermarked rows, adjusted for the number of watermarked rows. The formula is:

$$Z = \frac{\mu_{A_{\text{bit, W}}} - \mu_{A_{\text{bit, NW}}}}{\frac{\sigma_{A_{\text{bit, NW}}}}{\sqrt{n}}}$$

Where:

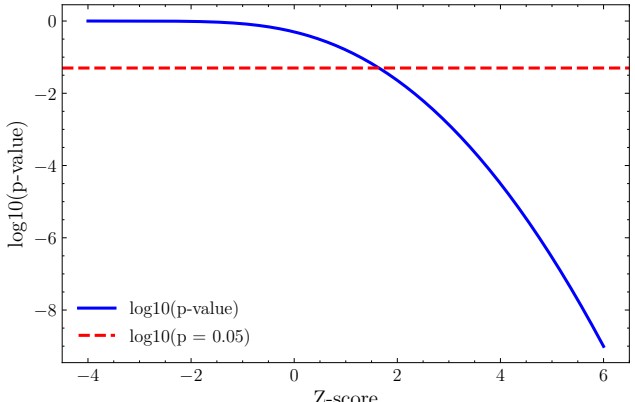

Figure 6: Relationship between Z-score and the logarithmic p-value for a one-tailed test. The red dashed line indicates the significance threshold ( $p = 0.05$, or $\log_{10}(p) \approx -1.3$ ).

- $\mu_{A_{\text{bit, W}}}$ is the mean bit accuracy of the watermarked rows,
- $\mu_{A_{\text{bit, NW}}}$ is the mean bit accuracy of the non-watermarked rows,
- $\sigma_{A_{\text{bit, NW}}}$ is the standard deviation of the bit accuracy in the non-watermarked rows,
- $n$ is the number of watermarked rows.

For the Tree-Ring method, the Z-score is computed based on the distance between the watermarked and non-watermarked data rather than row-wise accuracy. Since it's not row-wise, there is no dependence on $n$. Additionally, the Z-score is computed using the opposite tail of the distribution. The formula is:

$$Z = \frac{\mu_{\text{Dist}_{\text{NW}}} - \mu_{\text{Dist}_{\text{W}}}}{\sigma_{\text{Dist}_{\text{NW}}}}$$

Where:

- $\mu_{\text{Dist}_{\text{W}}}$ is the mean distance of the watermarked data,
- $\mu_{\text{Dist}_{\text{NW}}}$ is the mean distance of the non-watermarked data,
- $\sigma_{\text{Dist}_{\text{NW}}}$ is the standard deviation of the distance for the non-watermarked data.

The relationship between Z-score and the logarithmic p-value is shown in Figure 6.

- **TPR@XFPR**: This metric evaluates the True Positive Rate (TPR) at a given False Positive Rate (XFPR), where X is a predefined percentage. It provides insight into the watermark detection capability by measuring how often a watermarked table is correctly identified at different levels of false positives. A higher TPR@XFPR score indicates a better detection performance.

## F    ADDITIONAL RESULTS

### F.1    DISTRIBUTION OF LATENTS FOR DIFFERENT WATERMARKS

Figure 7 illustrates latent code matrices of size $(1000 \times 40)$ generated by Tree-Ring, Gaussian Shading, and `TabWak`. Upon observation, the latent codes produced by our method closely resemble a Gaussian distribution, whereas Tree-Ring and Gaussian Shading exhibit distinct patterns that are easily recognizable at first glance. The reduced distortion in the Gaussian distribution explains why our method achieves superior data quality.

### F.2    WATERMARKING NUMERICAL COLUMNS ONLY

Figure 8 presents TPR@1% FPR versus row count across four datasets. When applying the watermark only to numerical columns, the performance drops for the three watermarks. However, our

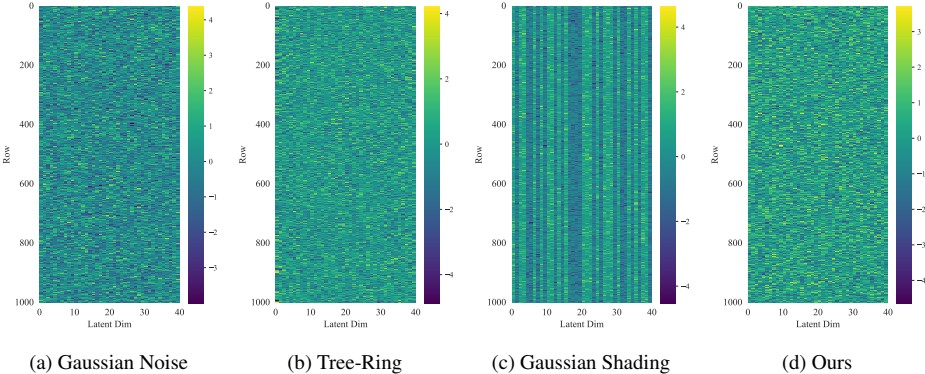

(a) Gaussian Noise      (b) Tree-Ring      (c) Gaussian Shading      (d) Ours

Figure 7: Examples of latent codes sampled from standard Gaussian noise and various watermarking methods

method still demonstrates strong robustness, achieving 100% TPR@1% FPR within 200 rows in 11 out of 12 cases.

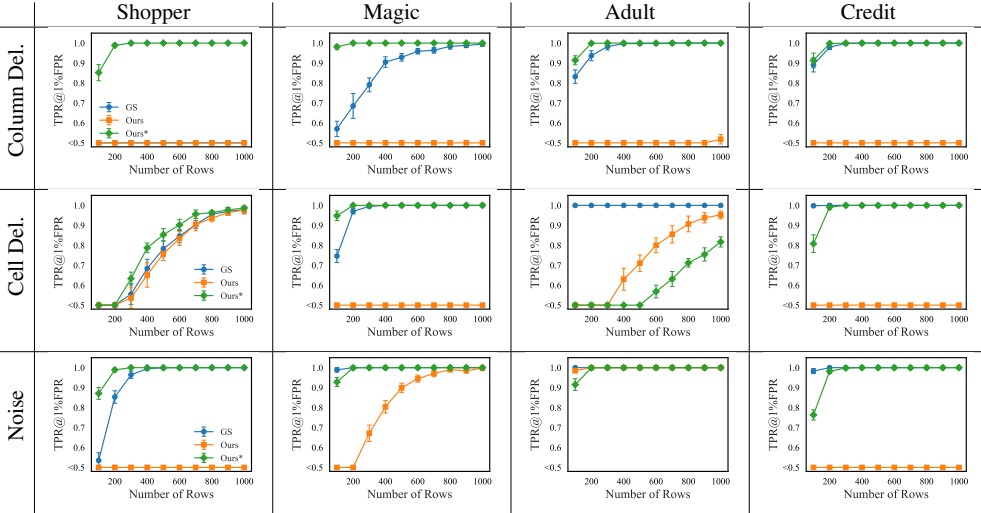

Figure 8: TPR@1% FPR versus row count in four datasets under various attacks, with the watermark applied only to the numeric columns Cell and Noise attacks are set at 20% strength; Column Deletion involves three columns.

### F.3 POST-PROCESSING WATERMARKS

The reason we did not include post-processing watermarks in the primary evaluation is that such methods, like the one described in (He et al., 2024), can only be embedded into continuous values by strategically adjusting these values to fall within a chosen range. However, the applicability of this approach is limited in tabular data, which often contains many integers and categorical values.

Additionally, post-processing watermarks are highly susceptible to common operations in tabular data processing, such as rounding, which can easily remove the watermark.

Since our dataset contains many integer columns, we preprocess the data by converting numbers into scientific notation. The method from (He et al., 2024), is then applied to the coefficients of the scientific notation. Below are the results of comparing the post-processing method under different types of attacks. The results show that the post-processing method is particularly vulnerable to Gaussian noise, where it fails to maintain the watermark.

| Dataset | Attacks | | | | | | | | | | | | |
|---------|---------|---|---|---|---|---|---|---|---|---|---|---|---|
| | Row Deletion | | | Column Deletion | | | Cell Deletion | | | Gaussian Noise | | | Shuffling |
| | 5% | 10% | 20% | 1 col | 2 col | 3 col | 5% | 10% | 20% | 5% | 10% | 20% | - |
| **Shoppers** | 65.3 | 63.5 | 59.9 | 67.1 | 67.1 | 67.1 | 63.3 | 60.0 | 53.3 | 0.1 | 0.0 | 0.0 | 67.1 |
| **Magic** | 38.4 | 37.3 | 35.3 | 39.3 | 39.1 | 39.3 | 37.3 | 35.2 | 31.9 | 0.0 | 0.1 | 0.0 | 39.4 |
| **Adult** | 68.9 | 67.1 | 63.2 | 70.7 | 70.7 | 70.7 | 67.0 | 63.3 | 56.3 | 0.0 | 0.0 | 0.6 | 70.7 |
| **Credit** | 62.2 | 60.5 | 57.2 | 63.8 | 63.7 | 63.7 | 61.2 | 58.2 | 51.4 | 0.0 | 0.0 | 0.0 | 63.8 |
| **Diabetes** | 56.3 | 54.8 | 51.6 | 57.8 | 57.8 | 57.8 | 54.7 | 51.9 | 45.9 | 0.0 | 0.0 | 0.0 | 57.8 |

Table 4: The robustness of post-processing watermark (He et al., 2024) against post-editing attacks: Average Z-score on 5K rows.

## F.4 LATENT-SPACE ATTACKS

In this subsection, we evaluate the robustness of various watermarking methods under latent-space manipulation attacks.

### F.4.1 REGENERATION ATTACK

The **Regeneration Attack** mimics the approach outlined in DiffPure (Nie et al., 2022). This attack maps the watermarked table into the latent space using the decoder inversion method, generating $\hat{\mathbf{z}}_0^W$. Subsequently, DDIM inversion transforms it into $\hat{\mathbf{z}}_T^W$, which serves as the initial latent for reconstructing the tabular data. The results in Table 5 reveal the impact of this attack on watermark detectability.

Sampling-based methods, including Tree-Ring, Gaussian Shading, and our method, retain their detectability under most of the regeneration process. However, Tree-Ring fails for the *Adult* dataset, demonstrating a limitation in its robustness. On the other hand, the post-processing watermark is entirely eliminated during regeneration, as evidenced by a Z-score of 0 across all datasets.

These findings underscore the vulnerability of post-processing watermarks to latent-space transformations, emphasizing the need for embedding mechanisms that are resilient to such attacks.

| Dataset | TR | GS | Ours | Ours* | He et al. (2024) |
|---------|-----|-----|------|-------|------------------|
| Shoppers | 4.73 | 22.30 | 11.02 | 35.10 | 0.00 |
| Magic | 4.85 | 36.53 | 13.68 | 25.09 | 0.21 |
| Adult | 0.54 | 46.42 | 42.08 | 30.50 | 0.01 |
| Credit | 5.34 | 84.06 | 10.86 | 22.13 | 0.07 |
| Diabetes | 6.29 | 55.76 | 5.82 | 7.04 | 0.03 |

Table 5: Robustness of different watermarking methods against the regeneration attack: Average Z-score on 5K rows.

### F.4.2 EMBEDDING ATTACK

The **Embedding Attack**, inspired by WAVES (An et al.), introduces adversarial perturbations to the numerical components of the watermarked table. Utilizing our encoder $\mathcal{E}$, which maps the tabular data $(X_{num}, X_{cat})$ to a latent representation, this attack generates perturbed data $X_{num}^{adv}$ that aims to shift the latent representation of the adversarial table away from that of the original watermarked table $X_{num}$. This objective is formulated as:

$$\max_{X_{num}^{adv}} \left\| \mathcal{E}(X_{num}^{adv}, X_{cat}) - \mathcal{E}(X_{num}, X_{cat}) \right\|_2,$$

subject to the constraint:

$$\left| X_{num}^{adv} - X_{num} \right| \leq \epsilon \cdot \left| X_{num} \right|.$$

In this formulation, $\epsilon$ is the perturbation budget, set to 0.2 in our experiments, ensuring that the modifications remain bounded while significantly impacting the latent representation.

The results in Table 6 highlight the effectiveness of this attack. Our method with the valid bit mechanism (Ours*) and Gaussian Shading demonstrate notable resilience, maintaining high Z-scores across most datasets. In contrast, the post-processing watermarking method is rendered ineffective, with Z-scores consistently approaching 0, signifying the complete destruction of the watermark.

This attack demonstrates the importance of designing embedding mechanisms that can withstand adversarial manipulations, ensuring the integrity and detectability of watermarks even under such challenging conditions.

| Dataset | TR | GS | Ours | Ours* | He et al. (2024) |
|---|---|---|---|---|---|
| Shoppers | 0.31 | 22.49 | 0.00 | 28.69 | 0.00 |
| Magic | 0.33 | 36.54 | 8.41 | 21.61 | 0.08 |
| Adult | 0.00 | 56.29 | 0.08 | 27.00 | 0.06 |
| Credit | 0.04 | 79.43 | 0.00 | 11.12 | 0.00 |
| Diabetes | 1.27 | 48.28 | 2.90 | 7.42 | 0.00 |

Table 6: Robustness of different watermarking methods against the embedding attack: Average Z-score on 5K rows.

### F.4.3 ADAPTIVE ATTACK ON TAIL VALUES IN LATENTS

To further challenge the watermarking methods, we propose an **Adaptive Attack** that targets the tail values in the latent space. This attack aims to minimize the contribution of outlier latent values ($\hat{z}_T$) while adhering to a perturbation constraint ($\epsilon = 0.2$). The optimization objective is formulated as:

$$\min_{X_{\text{num}}^{\text{adv}}} \left\| M_{\text{tail}} \cdot \hat{z}_T \right\|_2 ,$$

where the tail mask $M_{\text{tail}}$ is determined by the interquartile range of $\hat{z}_T$, defined as:

$$M_{\text{tail}}[i] = \begin{cases} 1 & \text{if } \hat{z}_T[i] < Q_{0.25}(\hat{z}_T) \text{ or } \hat{z}_T[i] > Q_{0.75}(\hat{z}_T), \\ 0 & \text{otherwise.} \end{cases}$$

The attack is subject to the constraint:

$$\left| X_{\text{num}}^{\text{adv}} - X_{\text{num}} \right| \leq \epsilon \cdot \left| X_{\text{num}} \right| .$$

In this framework, $\hat{z}_T$ represents the initial latent estimated using DDIM inversion applied to the encoder output of the perturbed tabular data. This approach ensures that the perturbations selectively target the tail values while maintaining bounded distortions to the original data.

The results of this attack, presented in Table 7, demonstrate that our method (Ours*) remains resilient under both Embedding and Adaptive Attacks, retaining Z-scores above 20 in most datasets. In contrast, the Adaptive Attack significantly reduces watermark robustness for weaker methods, such as Gaussian Shading, especially in datasets like *Magic* and *Diabetes*.

These findings highlight the importance of designing watermarking methods that leverage intrinsic latent-space properties to withstand targeted perturbations and ensure robust detectability under challenging conditions.

| Dataset | W/O Attack | Embedding Attack | Adaptive Attack |
|---|---|---|---|
| Shoppers | 34.52 | 28.69 | 24.61 |
| Magic | 25.30 | 21.61 | 7.43 |
| Adult | 28.45 | 27.00 | 26.03 |
| Credit | 22.91 | 11.12 | 14.48 |
| Diabetes | 7.86 | 7.42 | 2.15 |

Table 7: Robustness of TabWak against embedding and adaptive attacks: Average Z-score on 5K rows.

## F.5 HYPERPARAMETER EVALUATION FOR VALID BIT MECHANISM

Below are the experiments conducted to evaluate the hyperparameter settings of our method with the valid bit mechanism. We introduce a new setting for $l = 3$, where the standard normal distribution is divided into three quantiles. In this setting, we focus on the two tails: values $< \Phi^{-1}(0.333)$ and values $> \Phi^{-1}(0.667)$. The aim is to investigate whether the signs of the tail values differ in detecting self-cloning.

Tables 8 and 9 provide results for generative quality and robustness, respectively.

| Datasets | l | Shape | Trend | Logistic | MLE |
|---|---|---|---|---|---|
| **Shoppers** | W/O | 0.922 | 0.907 | 0.635 | 0.871 |
| | 3 | **0.908** | 0.893 | 0.567 | **0.879** |
| | 4 | 0.914 | **0.906** | **0.580** | 0.867 |
| **Magic** | W/O | 0.917 | 0.939 | 0.710 | 0.906 |
| | 3 | 0.903 | **0.936** | **0.736** | **0.893** |
| | 4 | **0.908** | 0.927 | 0.705 | 0.876 |
| **Adult** | W/O | 0.933 | 0.887 | 0.653 | 0.876 |
| | 3 | 0.927 | 0.867 | 0.636 | 0.871 |
| | 4 | **0.931** | **0.884** | **0.645** | **0.874** |
| **Credit** | W/O | 0.930 | 0.905 | 0.741 | 0.743 |
| | 3 | **0.927** | **0.897** | **0.713** | 0.741 |
| | 4 | 0.922 | 0.892 | 0.677 | **0.744** |
| **Diabetes** | W/O | 0.873 | 0.743 | 0.748 | 0.803 |
| | 3 | 0.832 | **0.735** | **0.728** | 0.789 |
| | 4 | **0.849** | 0.733 | 0.694 | **0.801** |

Table 8: Synthetic Table Quality: Comparison of hyperparameters $l = 3$ and $l = 4$. 'W/O' refers to data without watermark.

From Table 8, we observe that the quality results for $l = 3$ and $l = 4$ are close to each other. $l = 3$ achieves better performance in 10 out of 20 cases in the table (across different datasets and metrics).

| Dataset | l | Attacks | | | | | | | | | | | | |
|---|---|---|---|---|---|---|---|---|---|---|---|---|---|---|
| | | Row Deletion | | | Column Deletion | | | Cell Deletion | | | Gaussian Noise | | | Shuffling |
| | | 5% | 10% | 20% | 1 col | 2 col | 3 col | 5% | 10% | 20% | 5% | 10% | 20% | - |
| **Shoppers** | 3 | 29.06 | 28.33 | 26.57 | 29.82 | 30.10 | 31.49 | 28.55 | 27.92 | 26.52 | 24.49 | 28.03 | 39.11 | 29.79 |
| | 4 | **33.58** | **32.69** | **30.98** | **34.50** | **34.33** | **37.38** | **34.40** | **34.63** | **33.36** | **27.60** | **29.84** | **39.90** | **34.51** |
| **Magic** | 3 | 20.66 | 20.09 | 18.99 | 26.39 | 29.02 | 28.82 | 22.31 | 23.39 | 24.17 | 21.35 | 21.19 | 20.92 | 21.20 |
| | 4 | **24.78** | **23.98** | **22.61** | **32.38** | **32.33** | **37.80** | **26.92** | **28.13** | **30.17** | **25.51** | **25.12** | **25.06** | **25.39** |
| **Adult** | 3 | **31.18** | **30.37** | **28.64** | **32.21** | **31.70** | **28.74** | **31.76** | 29.95 | 28.25 | **37.34** | **54.67** | **69.21** | **32.01** |
| | 4 | 27.78 | 26.83 | 25.43 | 28.45 | 24.92 | 27.57 | 29.29 | **30.07** | **29.86** | 32.53 | 48.66 | 64.19 | 28.42 |
| **Credit** | 3 | 19.03 | 18.62 | 17.54 | 24.33 | 27.19 | 27.39 | 23.89 | 24.27 | 29.44 | 20.56 | 20.55 | 25.25 | 19.57 |
| | 4 | **22.11** | **21.65** | **20.29** | **27.31** | **32.71** | **34.98** | **26.65** | **30.31** | **36.24** | **23.18** | **24.31** | **27.17** | **22.88** |
| **Diabetes** | 3 | 5.75 | 5.62 | 5.29 | **9.97** | **13.14** | **15.94** | **6.89** | **7.07** | **6.60** | 5.13 | 4.23 | **4.11** | 5.73 |
| | 4 | **7.76** | **7.63** | **7.11** | 4.98 | 10.94 | 12.74 | 4.76 | 4.41 | 3.61 | **6.56** | **6.73** | 3.83 | **7.91** |

Table 9: Robustness of Different $l$ Settings of `TabWak` Against Post-Editing Attacks: Average Z-Score on 5K Rows.

From Table 9, we observe that $l = 4$ consistently achieves higher Z-scores than $l = 3$ in the Shoppers, Magic, and Credit datasets. In the Adult dataset, $l = 3$ performs better in 11 out of 13 cases, and in the Diabetes dataset, $l = 3$ wins in 7 out of 13 cases.

The better robustness of $l = 4$ can be attributed to valid bit values being closer to the distribution tails, making them more resistant to noise and distortion. However, increasing $l$ excessively may reduce robustness, as smaller quantile ranges introduce higher variance despite higher average bit accuracy. Excessively large $l$ values could also disrupt the initial latent distributions by imposing stricter constraints on self-cloning.

