# OpenReview forum: "TabWak: A Watermark for Tabular Diffusion Models"
_ICLR.cc/2025/Conference — ICLR 2025 Spotlight_

### Official Review · Reviewer_hNPH · 2024-11-01

**Soundness:** 3
**Presentation:** 3
**Contribution:** 3
**Rating:** 8
**Confidence:** 4

**Summary:**

This paper introduces a watermarking technique designed specifically for tabular diffusion models. The innovation lies in its row-wise watermarking approach that embeds signatures within the Gaussian latent codes used during the sampling process. Authors employ two mechanisms: a self-cloning plus shuffling technique that maintains row diversity while enabling row-level detection, and a valid-bit mechanism that leverages distribution tails for robust watermark detection. The method is tested across multiple datasets and demonstrates good performance in maintaining data utility while achieving reliable watermark detection.

**Strengths:**

The paper addresses an important gap in watermarking tabular data generated by diffusion models, which is becoming increasingly relevant as synthetic data adoption grows. Authors cleverly design self-cloning and shuffling mechanism to handle the challenges of tabular data, particularly the need for row-wise detection and inter-row diversity. And also provides solid theoretical analysis of bit accuracy expectations under Gaussian noise, both with and without the valid-bit mechanism. The experimental evaluation is thorough, covering multiple datasets, metrics, and attack scenarios.

**Weaknesses:**

1. Although the paper compares against adapted image watermarking techniques, it doesn't compare against existing tabular data watermarking methods that operate in post-processing (mentioned in Related Works, e.g., He et al., 2024). This comparison would help understand the relative advantages of embedding watermarks during sampling versus post-processing.
2. While the paper presents results with specific parameter choices (e.g., l=4 for quantiles), there's limited discussion about parameter sensitivity and how different choices might affect the trade-off between data quality and watermark robustness.
3. Although the paper mentions privacy protection as a use case for synthetic data, it doesn't analyze potential privacy implications of the watermarking scheme itself, such as whether the watermark could leak information about the training data or model architecture.

**Questions:**

1. How does TabWak's performance change with different architectures of the backbone diffusion model? The current evaluation uses a specific TabSyn architecture, would the method's effectiveness be maintained with other tabular diffusion model architectures?

2. Given that the valid-bit mechanism focuses on the tail of the distribution for improved robustness, could this create vulnerabilities to targeted attacks that specifically manipulate these tail values? Have you considered or evaluated such potential attacks?

---

> ### Author Response · Authors · 2024-11-18
> **Response to Reviewer hNPH (1)**
>
> Thank you for your valuable comments and positive feedback.
>
> **W1: Although the paper compares against adapted image watermarking techniques, it doesn't compare against existing tabular data watermarking methods that operate in post-processing (mentioned in Related Works, e.g., He et al., 2024). This comparison would help understand the relative advantages of embedding watermarks during sampling versus post-processing.**
>
> A: The reason we did not include post-processing watermarks is that such methods, like the one in [1], can only be embedded into continuous values by strategically adjusting these values to fall within a chosen range. However, the applicability of this approach is limited in tabular data, which often contains many integers and categorical values. Additionally, post-processing watermarks are highly susceptible to common operations in tabular data processing, such as rounding, which can easily remove the watermark.
>
> Since our dataset contains many integer columns, we first convert the numbers into scientific notation during preprocessing. The method from [1] is then applied to the coefficients of the scientific notation. Below are the results of comparing the post-processing method under different types of attacks. The results show that the post-processing method is particularly vulnerable to Gaussian noise, where it  fails to maintain the watermark.
>
> | Dataset   | Row Deletion |      |      | Column Deletion |       |       | Cell Deletion |      |      | Gaussian Noise |     |     | Shuffling |
> |-----------|--------------|------|------|-----------------|-------|-------|---------------|------|------|----------------|-----|-----|-----------|
> |           | 5%           | 10%  | 20%  | 1 col           | 2 col | 3 col | 5%            | 10%  | 20%  | 5%             | 10% | 20% |           |
> | Shoppers  | 65.3         | 63.5 | 59.9 | 67.1            | 67.1  | 67.1  | 63.3          | 60.0 | 53.3 | 0.1            | 0.0 | 0.0 | 67.1      |
> | Magic     | 38.4         | 37.3 | 35.3 | 39.3            | 39.1  | 39.3  | 37.3          | 35.2 | 31.9 | 0.0            | 0.1 | 0.0 | 39.4      |
> | Adult     | 68.9         | 67.1 | 63.2 | 70.7            | 70.7  | 70.7  | 67.0          | 63.3 | 56.3 | 0.0            | 0.0 | 0.6 | 70.7      |
> | Credit    | 62.2         | 60.5 | 57.2 | 63.8            | 63.7  | 63.7  | 61.2          | 58.2 | 51.4 | 0.0            | 0.0 | 0.0 | 63.8      |
> | Diiabetes | 56.3         | 54.8 | 51.6 | 57.8            | 57.8  | 57.8  | 54.7          | 51.9 | 45.9 | 0.0            | 0.0 | 0.0 | 57.8      |
>
> **Table A. The Robustness of  Post-processing Watermark [1] Against Post-Editing Attacks: Average Z-score on 5K rows**

---

> > ### Author Response · Authors · 2024-11-18
> > **Response to Reviewer hNPH (2)**
> >
> > We also designed two new attacks to manipulate the latent space based on feedback from other reviewers. The first attack, referred to as the **Regeneration Attack**, is inspired by the approach in [2]. In this attack, we employ the decoder inversion described in our paper to map the watermarked table to the latent representation, denoted as $\hat{\mathbf{z}}_0^W$. Subsequently, we use DDIM inversion to generate $\hat{\mathbf{z}}_T^W$. This newly generated initial latent representation is then used to reconstruct the tabular data. The regenerated tabular data is evaluated using the watermarking method. The results of the regeneration attack are presented in the table below.
> >
> > | Dataset  | TR   | GS    | Ours  | Ours\* | Post-processing |
> > | -------- | ---- | ----- | ----- | ------ | --------------- |
> > | Shoppers | 4.73 | 22.30 | 11.02 | 35.10  | 0.00            |
> > | Magic    | 4.85 | 36.53 | 13.68 | 25.09  | 0.21            |
> > | Adult    | 0.54 | 46.42 | 42.08 | 30.50  | 0.01            |
> > | Credit   | 5.34 | 84.06 | 10.86 | 22.13  | 0.07            |
> > | Diabetes | 6.29 | 55.76 | 5.82  | 7.04   | 0.03            |
> >
> > **Table B. Robustness of different watermarking methods against the regeneration attack: Average Z-score on 5K rows**
> >
> > From the results, we observe that during the regeneration process, the watermarks for sampling-based methods remain detectable (except for TR on the Adult dataset, where it fails even without the attack). In contrast, the watermark applied through post-processing is entirely removed during regeneration.
> >
> > The second attack, referred to as the **Embedding Attack**, is inspired by WAVES [3]. Utilizing our encoder $\mathcal{E}$, which maps the tabular data $(X_{num}, X_{cat})$ to a latent representation, we introduce perturbations to the numerical component of the tabular data, denoted as $X_{num}^{adv}$. These perturbations aim to deviate the latent representation of the adversarial table from that of the original watermarked table, $X_{num}$, while staying within a perturbation constraint. Formally, this is expressed as:
> >
> > $$
> > \max \left\|\mathcal{E}(X_{num}^{adv}, X_{cat}) - \mathcal{E}(X_{num}, X_{cat})\right\|_2,
> > $$
> >
> > subject to the constraint
> >
> > $$
> > \left|X_{n u m}^{a d v}-X_{n u m}\right| \leq \epsilon \cdot\left|X_{n u m}\right|
> > $$
> >
> > In our setting, $\epsilon$ is set to 0.2. The results are presented in the table below.
> >
> > | Dataset  | TR   | GS    | Ours | Ours\* | Post-processing |
> > | -------- | ---- | ----- | ---- | ------ | --------------- |
> > | shoppers | 0.31 | 22.49 | 0.00 | 28.69  | 0.00            |
> > | magic    | 0.33 | 36.54 | 8.41 | 21.61  | 0.08            |
> > | adult    | 0.00    | 56.29 | 0.08 | 27.00  | 0.06            |
> > | credit   | 0.04 | 79.43 | 0.00 | 11.12  | 0.00            |
> > | diabetes | 1.27 | 48.28 | 2.90 | 7.42   | 0.00            |
> >
> >
> > **Table C. The Robustness of different watermarking methods Against Embedding Attack: Average Z-score on 5K rows**
> >
> > From the results, we observe that in this attack setting, our method with the valid bit mechanism (Ours*) and Gaussian Shading (GS) demonstrates strong robustness against attacks. In contrast, the post-processing watermark fails across all datasets, as the perturbation completely destroys the watermark.

---

> ### Author Response · Authors · 2024-11-18
> **Response to Reviewer hNPH (3)**
>
> **W2: While the paper presents results with specific parameter choices (e.g., l=4 for quantiles), there's limited discussion about parameter sensitivity and how different choices might affect the trade-off between data quality and watermark robustness.**
>
> A: We acknowledge that the discussion about the sensitivity of $l$ is limited. However, we believe that the choice of $l$ has minimal impact on data quality if $l$ is not too large. Regardless of the value of $l$, our method samples the initial latent from each quantile of the Gaussian distribution with probabilities proportional to each quantile. Regarding robustness, we agree that further exploration of hyperparameters could provide valuable insights. We are testing the different settings $l=3$ now. We will update you when we get results.
>
> **W3: Although the paper mentions privacy protection as a use case for synthetic data, it doesn't analyze potential privacy implications of the watermarking scheme itself, such as whether the watermark could leak information about the training data or model architecture.**
>
> A: For sampling-based methods, we believe it is more difficult to leak information about the training data or model architecture. This is because the only modification introduced during watermarking is controlling the sampling process from the Gaussian distribution for the initial latent $\mathbf{z}_T$. Even if an adversary gains access to both the model and the watermarking method, they would only be able to recover the tabular data back to its initial latent representation.
> Furthermore, our method ensures that the initial latent $\mathbf{z}_T$ remains close to the Gaussian distribution, as demonstrated in Figure 6 of the Appendix. This makes it challenging for an adversary to uncover any hidden patterns in our sampling process, even if they successfully recover the initial latent.
>
> **Q1: How does TabWak's performance change with different architectures of the backbone diffusion model? The current evaluation uses a specific TabSyn architecture, would the method's effectiveness be maintained with other tabular diffusion model architectures?**
>
> A: We believe our method is adaptable to various backbone diffusion models. Similar to techniques like Tree-Ring, which have demonstrated consistent performance across different diffusion model architectures, our approach does not rely on any modules specifically tailored to a particular model architecture. Therefore, we expect its effectiveness to generalize well across other architectures.
>
> **Q2: Given that the valid-bit mechanism focuses on the tail of the distribution for improved robustness, could this create vulnerabilities to targeted attacks that specifically manipulate these tail values? Have you considered or evaluated such potential attacks?**
>
> A: As discussed in our response to W3, it is inherently challenging for an adversary to access the initial latent representation directly, making it difficult to manipulate. However, targeted attacks could theoretically be developed where the attacker reconstructs the initial latent representation and selectively alters the tail values. For instance, they might shift these tail values toward the distribution center or invert their signs. We are working on it right now.
>
> ### References
> [1] He, Hengzhi, et al. "Watermarking generative tabular data." arXiv preprint arXiv:2405.14018 (2024)
>
> [2] Nie, Weili, et al. "Diffusion Models for Adversarial Purification." International Conference on Machine Learning. PMLR, 2022.
>
> [3] An, Bang, et al. "Benchmarking the robustness of image watermarks." arXiv preprint arXiv:2401.08573 (2024).

---

> > ### Author Response · Authors · 2024-11-22
> > **Response to Reviewer hNPH (4) (Supplement to W2)**
> >
> > **Dear Reviewer hNPH,**
> >
> > Below are the experiments conducted to evaluate the hyperparameter settings of our method with the valid bit mechanism. We introduce a new setting for $l = 3$, where the standard normal distribution is divided into three quantiles. In this setting, we focus on the two tails: values $< \Phi^{-1}(0.333)$ and values $> \Phi^{-1}(0.667)$. The aim is to investigate whether the signs of the tail values differ in detecting self-cloning.
> >
> > Table D and Table E provide results for generative quality and robustness, respectively.
> >
> > | **Datasets** | **l**   | **Shape** | **Trend** | **Logistic** | **MLE**   |
> > |--------------|---------|-----------|-----------|--------------|-----------|
> > | **Shoppers** | W/O     | 0.922     | 0.907     | 0.635        | 0.871     |
> > |              | 3       | **0.908** | 0.893     | 0.567        | **0.879** |
> > |              | 4       | 0.914     | **0.906** | **0.580**    | 0.867     |
> > | **Magic**    | W/O     | 0.917     | 0.939     | 0.710        | 0.906     |
> > |              | 3       | 0.903     | **0.936** | **0.736**    | **0.893** |
> > |              | 4       | **0.908** | 0.927     | 0.705        | 0.876     |
> > | **Adult**    | W/O     | 0.933     | 0.887     | 0.653        | 0.876     |
> > |              | 3       | 0.927     | 0.867     | 0.636        | 0.871     |
> > |              | 4       | **0.931** | **0.884** | **0.645**    | **0.874** |
> > | **Credit**   | W/O     | 0.930     | 0.905     | 0.741        | 0.743     |
> > |              | 3       | **0.927** | **0.897** | **0.713**    | 0.741     |
> > |              | 4       | 0.922     | 0.892     | 0.677        | **0.744** |
> > | **Diabetes** | W/O     | 0.873     | 0.743     | 0.748        | 0.803     |
> > |              | 3       | 0.832     | **0.735** | **0.728**    | 0.789     |
> > |              | 4       | **0.849** | 0.733     | 0.694        | **0.801** |
> >
> > **Table D. Synthetic Table Quality: Comparison of hyperparameters $l=3$ and $l=4$. `W/O` refers to data without watermark.**
> >
> > From Table D, we observe that the quality results for $l=3$ and $l=4$ are close to each other. $l=3$ achieves a better performance in 10 out of 20 cases in the table (across different datasets and metrics).

---

> > > ### Author Response · Authors · 2024-11-22
> > > **Response to Reviewer hNPH (5) (Supplement to W2)**
> > >
> > > | **Dataset**  | **l** | **Row Deletion** |           |           | **Column Deletion** |           |           | **Cell Deletion** |           |           | **Gaussian Noise** |           |           | **Shuffling**  |
> > > |--------------|-------|------------------|-----------|-----------|---------------------|-----------|-----------|-------------------|-----------|-----------|--------------------|-----------|-----------|----------------|
> > > |              |       | **5%**           | **10%**   | **20%**   | **1 col**           | **2 col** | **3 col** | **5%**            | **10%**   | **20%**   | **5%**            | **10%**   | **20%**   |                |
> > > | **Shoppers** | 3     | 29.06            | 28.33     | 26.57     | 29.82               | 30.10     | 31.49     | 28.55             | 27.92     | 26.52     | 24.49             | 28.03     | 39.11     | 29.79          |
> > > |              | 4     | **33.58**        | **32.69** | **30.98** | **34.50**           | **34.33** | **37.38** | **34.40**         | **34.63** | **33.36** | **27.60**         | **29.84** | **39.90** | **34.51**      |
> > > | **Magic**    | 3     | 20.66            | 20.09     | 18.99     | 26.39               | 29.02     | 28.82     | 22.31             | 23.39     | 24.17     | 21.35             | 21.19     | 20.92     | 21.20          |
> > > |              | 4     | **24.78**        | **23.98** | **22.61** | **32.38**           | **32.33** | **37.80** | **26.92**         | **28.13** | **30.17** | **25.51**         | **25.12** | **25.06** | **25.39**      |
> > > | **Adult**    | 3     | **31.18**        | **30.37** | **28.64** | **32.21**           | **31.70** | **28.74** | **31.76**         | 29.95     | 28.25     | **37.34**         | **54.67** | **69.21** | **32.01**      |
> > > |              | 4     | 27.78            | 26.83     | 25.43     | 28.45               | 24.92     | 27.57     | 29.29             | **30.07** | **29.86** | 32.53             | 48.66     | 64.19     | 28.42          |
> > > | **Credit**   | 3     | 19.03            | 18.62     | 17.54     | 24.33               | 27.19     | 27.39     | 23.89             | 24.27     | 29.44     | 20.56             | 20.55     | 25.25     | 19.57          |
> > > |              | 4     | **22.11**        | **21.65** | **20.29** | **27.31**           | **32.71** | **34.98** | **26.65**         | **30.31** | **36.24** | **23.18**         | **24.31** | **27.17** | **22.88**      |
> > > | **Diabetes** | 3     | 5.75             | 5.62      | 5.29      | **9.97**            | **13.14** | **15.94** | **6.89**          | **7.07**  | **6.60**  | 5.13              | 4.23      | **4.11**  | 5.73           |
> > > |              | 4     | **7.76**         | **7.63**  | **7.11**  | 4.98                | 10.94     | 12.74     | 4.76              | 4.41      | 3.61      | **6.56**          | **6.73**  | 3.83      | **7.91**       |
> > >
> > > **Table E. Robustness of Different $l$ Settings of TabWak Against Post-Editing Attacks: Average Z-Score on 5K Rows**
> > >
> > > From Table E, we observe that $l=4$ consistently achieves higher Z-scores than $l=3$ in the Shoppers, Magic, and Credit datasets. In the Adult dataset, $l=3$ performs better in 11 out of 13 cases, and in the Diabetes dataset, $l=3$ wins in 7 out of 13 cases.
> > >
> > > The better robustness of $l=4$ can be attributed to valid bit values being closer to the distribution tails, making them more resistant to noise and distortion. However, increasing $l$ excessively may reduce robustness, as smaller quantile ranges introduce higher variance despite higher average bit accuracy. Excessively large $l$ values could also disrupt the initial latent distributions by imposing stricter constraints on self-cloning.

---

> ### Author Response · Authors · 2024-11-24
> **Response to Reviewer hNPH (6) (Supplement to Q2)**
>
> Based on your suggestions, we developed an adaptive attack that focuses on the tail values in the original latent. We additionally introduce perturbations to the numerical values in the table, denoted as $X_{\text{num}}$. Specifically, we use the encoder to approximate the latent $\hat{z}_0$, then employ the diffusion model for DDIM inversion to estimate the initial latent $\hat{z}_T$. Our attack minimizes the tail values of the latent $\hat{z}_T$, which can be formally expressed as:
>
> $$
> \min_{X_{\mathrm{num}}^{\mathrm{adv}}} \left\| M_{\mathrm{tail}} \cdot \hat{z}_T \right\|_2,
> $$
>
> subject to the constraint:
>
> $$
> \left| X_{\mathrm{num}}^{\mathrm{adv}} - X_{\mathrm{num}} \right| \leq \epsilon \cdot \left| X_{\mathrm{num}} \right|,
> $$
>
> where
>
> $$
> \hat{z}_T = DDIM^{-1}(\mathcal{E}(X_{\mathrm{num}}^{\mathrm{adv}})),
> $$
>
> and
>
> $$
> M_{\mathrm{tail}}[i] =
> \begin{cases}
> 1 & \text{if } \hat{z}_T[i] < Q_{0.25}(\hat{z}_T) \text{ or } \hat{z}_T[i] > Q_{0.75}(\hat{z}_T), \\
> 0 & \text{otherwise.}
> \end{cases}
> $$
>
>
> In our experiments, $\epsilon$ is set to 0.2. For DDIM inversion, we limit the number of steps to 10 to accelerate the process and reduce backpropagation overhead during optimization. The final results are summarized below:
>
> | Dataset  | W/O Attack   | Embedding Attack    | Adaptive Attack |
> | -------- | ------------ | ------------------- | ---------------- |
> | Shoppers | 34.52        | 28.69              | 24.61            |
> | Magic    | 25.30        | 21.61              | 7.43             |
> | Adult    | 28.45        | 27.00              | 26.03            |
> | Credit   | 22.91        | 11.12              | 14.48            |
> | Diabetes | 7.86         | 7.42               | 2.15             |
>
> **Table F. The Robustness of TabWak Against Embedding and Adaptive Attacks: Average Z-score on 5K rows**
>
> From the results, we observe that, under the same $\epsilon$, adaptive attacks reduced the average Z-score more significantly in 4 out of 5 datasets. Notably, the attack was particularly successful on the `magic` and `diabetes` datasets. We attribute this to the fact that we only perturbed numerical columns in the tabular data. In these two datasets, all columns are numerical except for the target columns, while the other three datasets contain more categorical columns. We will include these additional results in the appendix.

---

> > ### Comment · Reviewer_hNPH · 2024-11-25
> >
> > Thanks for your responses, all of my concerns have been addressed. I have adjusted the score.

---

> > > ### Author Response · Authors · 2024-11-25
> > >
> > > Reviewer hNPH,
> > >
> > > Thank you for your encouraging feedback. We are delighted that our detailed response and revisions have met your expectations. Your insightful suggestions have significantly improved our manuscript, and we truly value your effort and guidance.

---

### Official Review · Reviewer_jg7M · 2024-11-01

**Soundness:** 3
**Presentation:** 3
**Contribution:** 2
**Rating:** 6
**Confidence:** 3

**Summary:**

In this paper, the authors propose a novel watermarking method for tabular diffusion models. As the first sampling-phase watermark for tabular data, this method controls the initial seed of the latent diffusion model using symmetric seeds, with bitwise accuracy between two halves applied at the detection phase. Additionally, the initial noise for each row varies to ensure a diverse generation. In experiments, the authors show the proposed method is robust against various attacks.

**Strengths:**

- The paper is well-written and easy to follow and understand.
- The method is very interesting. Although it is kind of inspired by the previous work, the new design like self-cloning is very interesting and insightful.
- The results are very promising. The method can achieve really good generation quality while the detectability is also good.
- The paper has a very solid evaluation. Especially, the authors use 4 quality metrics to thoroughly show the diversity and usefulness of the generated data.

**Weaknesses:**

- I think my main concern is the usefulness of tabular data watermarking. I can see for images or texts, we need to detect if they are AI-generated to prevent spreading misinformation, but I don't see tabular data can bring much harm like other modalities.
- In terms of detectability, the proposed method underperforms the Gaussian Shading baseline by a lot in most of the datasets.
- It will be very helpful to include other tabular watermark baselines even though they are not sampling-phase methods.

**Questions:**

- The proposed method is very good for Shoppers, but Gaussian Shading has way higher z-scores in other datasets. Do the authors have any idea why it's the case? Also, I guess you can sacrifice the generation quality to make the detectability much better.
- I think the proposed method is very interesting. Do the authors know if a similar method can be applied to images?

---

> ### Author Response · Authors · 2024-11-18
> **Response to Reviewer jg7M (1)**
>
> Thank you for your valuable comments.
>
> **W1: I think my main concern is the usefulness of tabular data watermarking. I can see for images or texts, we need to detect if they are AI-generated to prevent spreading misinformation, but I don't see tabular data can bring much harm like other modalities.**
>
> A: Yes, text and images are often prioritized due to their visible influence on daily life. However, synthetic tabular is the most common modality in industries and organizations, which increasingly embrace synthetic data as a privacy-preserving data-sharing solution [1-3]. It is important for the synthetic data generator to verify if a piece of table is generated by itself and then take responsibility for the (misa)usage of such data. Synthetic tables pose subtle yet significant risks.
> For instance: 1) Financial Fraud: Synthetic datasets can manipulate performance metrics, enabling hedge funds to fabricate high returns and conceal losses. Watermarking ensures that only genuine data is used for informed decision-making. 2) Healthcare Misdiagnosis: Altered synthetic patient data can skew diagnostic tools or treatment recommendations, potentially leading to issues like over-prescription of medications. Watermarking safeguards data integrity, fostering trust in healthcare models. 3) Regulatory Evasion: Companies may exploit synthetic data to falsify compliance records, inflate profits, or create misleading sustainability reports.
>
> Watermarking confirms data authenticity, ensuring reliability in audits. The watermarking technique can also protect the copyright of generated tabular data for the model owner, ensuring that the data's ownership and intellectual property rights are safeguarded by the model itself.

---

> ### Author Response · Authors · 2024-11-18
> **Response to Reviewer jg7M (2)**
>
> **W2 & Q1: In terms of detectability, the proposed method underperforms the Gaussian Shading baseline by a lot in most of the datasets. The proposed method is very good for Shoppers, but Gaussian Shading has way higher z-scores in other datasets. Do the authors have any idea why it's the case? Also, I guess you can sacrifice the generation quality to make the detectability much better.**
>
> A: Indeed, compared to the Z-score, Gaussian Shading exhibits superior robustness. However, when considering the absolute value of the p-value, our proposed method also performs well. The key to Gaussian Shading's better performance lies in its row-by-row detection approach: it employs the same latent seed for each row during generation, making detection significantly easier. In contrast, TabWak uses different latent seeds for each row, which constrains the fidelity of the latent space and leads to uneven distribution, ultimately affecting generation quality. Despite this, we believe that TabWak achieves a more optimal balance between robustness and data quality. Our reasoning is as follows:
>
> 1. Detectability in Terms of p-value: While the Z-scores in Table 2 for our method (Our*) are indeed lower, the detectability of our method remains competitive. Even in the worst-case scenario for our method, the Z-score ( Diabetes dataset under 20% cell deletion attacks) corresponds to a p-value of 1.5e-4. This implies that even under the strongest attack (5k rows), our method maintains a low false positive rate of 1.5e-4. This shows that the detectability of our method is still reliable, despite its slightly lower robustness.
>
> 2. Trade-off with Data Quality: In contrast, Gaussian Shading, while exhibiting superior robustness, significantly compromises data quality. This is due to its reliance on using the same latent seed across all rows, which introduces noticeable patterns in the watermarked data. Our method avoids such performance drop, preserving the overall quality of the dataset better than Gaussian Shading.
>
> To illustrate this trade-off, we present the following figures (hosted anonymously), showing the relationship between p-value under various attacks and average data quality ( (from Table 1, specifically the average of `Shape`, `Trend`, `Logistic`, `MLE`). Notably, our method (Ours*, represented by filled plus markers) predominantly occupies the upper-left region of the plots, reflecting superior performance across most scenarios, except under cell deletion attacks and Gaussian noise attacks on the Diabetes dataset. On the other hand, Gaussian Shading (GS), while robust in detectability, consistently falls in the lower-left region, emphasizing its trade-off of reduced data quality for robustness.
>
> [Figure A. Trade-off Analysis: Quality and Robustness Under 20% Row Deletion](https://postimg.cc/5X1YnYSP)
>
> [Figure B. Trade-off Analysis: Quality and Robustness Under 3-Column Deletion](https://postimg.cc/Yv34VrXt)
>
> [Figure C. Trade-off Analysis: Quality and Robustness Under 20% Cell Deletion](https://postimg.cc/QV2WKn9Z)
>
> [Figure D. Trade-off Analysis: Quality and Robustness Under 20% Gaussian Noise](https://postimg.cc/crftRMwg)
>
>
> And for the question if we can sacrifice the generation quality to make the detectability much better. Regarding the possibility of sacrificing generation quality to achieve better detectability, we propose that the latent distributions for multiple rows remain close to a Gaussian distribution. While robustness could be further enhanced by distorting the distribution— e.g., adding bias to increase values in the tails—we believe the current trade-off is well-balanced, as evidenced in the results above.

---

> ### Author Response · Authors · 2024-11-18
> **Response to Reviewer jg7M (3)**
>
> **W3: It will be very helpful to include other tabular watermark baselines even though they are not sampling-phase methods.**
>
> A: The reason we did not include post-processing watermarks is that such methods, like the one in [4], can only be embedded into continuous values by strategically adjusting these values to fall within a chosen range. However, the applicability of this approach is limited in tabular data, which often contains many integers and categorical values. Additionally, post-processing watermarks are highly susceptible to common operations in tabular data processing, such as rounding, which can easily remove the watermark.
>
> Since our dataset contains many integer columns, we first convert the numbers into scientific notation during preprocessing. The method from [4] is then applied to the coefficients of the scientific notation. Below are the results of comparing the post-processing method under different types of attacks. The results show that the post-processing method is particularly vulnerable to Gaussian noise, where it often fails to maintain the watermark.
>
> | Dataset   | Row Deletion |      |      | Column Deletion |       |       | Cell Deletion |      |      | Gaussian Noise |     |     | Shuffling |
> |-----------|--------------|------|------|-----------------|-------|-------|---------------|------|------|----------------|-----|-----|-----------|
> |           | 5%           | 10%  | 20%  | 1 col           | 2 col | 3 col | 5%            | 10%  | 20%  | 5%             | 10% | 20% |           |
> | Shoppers  | 65.3         | 63.5 | 59.9 | 67.1            | 67.1  | 67.1  | 63.3          | 60.0 | 53.3 | 0.1            | 0.0 | 0.0 | 67.1      |
> | Magic     | 38.4         | 37.3 | 35.3 | 39.3            | 39.1  | 39.3  | 37.3          | 35.2 | 31.9 | 0.0            | 0.1 | 0.0 | 39.4      |
> | Adult     | 68.9         | 67.1 | 63.2 | 70.7            | 70.7  | 70.7  | 67.0          | 63.3 | 56.3 | 0.0            | 0.0 | 0.6 | 70.7      |
> | Credit    | 62.2         | 60.5 | 57.2 | 63.8            | 63.7  | 63.7  | 61.2          | 58.2 | 51.4 | 0.0            | 0.0 | 0.0 | 63.8      |
> | Diiabetes | 56.3         | 54.8 | 51.6 | 57.8            | 57.8  | 57.8  | 54.7          | 51.9 | 45.9 | 0.0            | 0.0 | 0.0 | 57.8      |
>
> **Table A. The Robustness of  Post-processing Watermark [4] Against Post-Editing Attacks: Average Z-score on 5K rows**
>
> **Q2: Do the authors know if a similar method can be applied to images?**
>
> A:  Our method aims to improve the quality of tabular data by avoiding the reuse of the same latent seed for each row. This ensures greater diversity in the latent representations, ultimately enhancing the quality of the generated table. In tabular data generation, each row is analogous to an individual image in image generation. However, in image generation, it is less critical to use different latent seeds for each image since evaluation typically does not involve comparisons across images, and the latent space for images is much larger than that for a single row of tabular data. But we believe that if the task involves generating a batch of images using the same text prompt, our method could also be beneficial for improving the diversity within the batch.
>
> ### References
> [1] Liu, Fan, et al. "Privacy-preserving synthetic data generation for recommendation systems." Proceedings of the 45th International ACM SIGIR Conference on Research and Development in Information Retrieval. 2022.
>
> [2] Qian, Zhaozhi, et al. "Synthetic data for privacy-preserving clinical risk prediction." Scientific Reports 14.1 (2024): 25676.
>
> [3] Potluru, Vamsi K., et al. "Synthetic data applications in finance." arXiv preprint arXiv:2401.00081 (2023).
>
> [4] He, Hengzhi, et al. "Watermarking generative tabular data." arXiv preprint arXiv:2405.14018 (2024)

---

> > ### Comment · Reviewer_jg7M · 2024-11-22
> > **Thank you for your response**
> >
> > I really appreciate the authors' detailed response, which addresses my concerns. Therefore, I have increased my score to positive accordingly.

---

> > > ### Author Response · Authors · 2024-11-24
> > > **Official Comment by Authors**
> > >
> > > Dear Reviewer jg7M,
> > >
> > > We are delighted to see that our detailed response addressed your initial concerns and that both our work and rebuttal have met your expectations. Your constructive suggestions have been instrumental in improving the quality of our manuscript, and we deeply appreciate your thoughtful review.

---

### Official Review · Reviewer_CBoB · 2024-11-03

**Soundness:** 3
**Presentation:** 3
**Contribution:** 2
**Rating:** 6
**Confidence:** 4

**Summary:**

This manuscript introduces a novel watermarking method for tabular diffusion models, claiming to be the first to embed invisible signatures that control Gaussian latent codes for synthesizing table rows. Image-based watermarking methods are unsuitable as they impair row-wise watermark detection, row diversity, and robustness against tabular-specific attacks like row deletion and shuffling. To address this, the authors propose a self-cloning plus shuffling mechanism and a valid bit mechanism to enhance watermark robustness and table quality. Extensive experiments on five datasets demonstrate the method's effectiveness compared to SOTA diffusion watermarking approaches.

**Strengths:**

* This manuscript makes a valuable first attempt at watermarking tabular diffusion models by embedding invisible signatures that control Gaussian latent code sampling to synthesize table rows via a diffusion backbone.

* The authors effectively identify key differences between tabular and other diffusion models, recognizing the unique distortions that tables face. They design a self-cloning plus shuffling mechanism and a valid bit mechanism to enhance both the quality of watermarked tables and the robustness of the embedded watermark, demonstrating a strong understanding of the practical demands of tabular diffusion models.

* To validate their method, the authors provide both thorough theoretical support and a substantial set of experiments, underscoring the rigor of their approach, which is commendable.

* The manuscript is well-written, with a clear explanation of the authors’ thought process and design steps, facilitating readers' understanding of this work.

**Weaknesses:**

* The watermarking method for tabular diffusion models in this manuscript appears to rely heavily on the previous diffusion model watermarking approach [1]. While some new components and adaptations address the specific needs of tabular data, the novelty could be further emphasized by exploring unique features and applications of table data in greater depth. Expanding on these aspects would enhance the distinctiveness of the contribution.

* In designing the "valid bit mechanism," the authors focus on extrema values at the ends of the distribution, which is logical but may underutilize the central parts of the distribution. To validate this choice, theoretical proof or empirical evidence is needed to show that this approach outperforms one that also uses centered values for repeated information, as repetition might improve robustness.

* The experimental results are not fully satisfactory. For instance, in Table 2, **the robustness of the proposed method falls short of [1] across four out of five datasets.** The explanations for this gap are limited and may not fully justify the proposed method's advantages. Additional experimental data should be provided to demonstrate the method’s superiority, or the authors should offer a more comprehensive analysis of the results, clarifying how the proposed method remains advantageous despite the observed performance gap.

[1]. Gaussian shading: Provable performance-lossless image watermarking for diffusion models. In Proceedings of the IEEE/CVF Conference on Computer Vision and Pattern Recognition, pp. 12162–12171, 2024.

**Questions:**

* How do the authors plan to enhance the novelty of their work by incorporating more innovative contributions specific to tabular diffusion models, rather than relying on previous achievements? Could they explore further applications or unique features of tabular data to better distinguish this approach?

* Could the authors provide theoretical or experimental comparisons between the "valid bit mechanism" and alternative approaches, such as information repetition, to clarify its effectiveness and superiority?

* Given that the current experimental results show the proposed method’s performance is not consistently superior to previous watermarking methods, can the authors provide additional data or a more detailed explanation of these outcomes? How do they address this limitation to demonstrate the proposed method's advantages more clearly?

---

> ### Author Response · Authors · 2024-11-18
> **Response to Reviewer CBoB (1)**
>
> Thank you for your valuable comments. Below, we provide our responses to the weaknesses and questions you highlighted.
>
> **W1&A1: This manuscript's watermarking method for tabular diffusion models builds on a previous diffusion model watermarking approach [1]. While some adaptations address tabular data's specific needs, the novelty could be emphasized further by exploring unique features and applications of table data. How do the authors plan to enhance the distinctiveness of their approach beyond previous achievements? Could additional innovations specific to tabular data be incorporated?**
>
> A: While our method builds on previous work, its novelty lies in our specific adaptations,  which address the row-by-row watermarking and detection challenges unique to tabular data through self-cloning and valid-bit mechanisms. Additionally, we are the first to introduce watermarking during the sampling phase for tabular data, embedding the watermark in the latent space. This makes it harder to identify and applicable to both continuous and categorical columns, unlike post-processing methods like [2], which are limited to continuous columns. Regarding the unique properties of tabular data, we have begun to consider the challenges posed by categorical columns, such as the sparsity introduced by one-hot encoding and the risk of losing watermark information due to the non-differentiability of the arg max operation. We recognize these as important issues and plan to address them in future work while continuing to explore other distinctive aspects of tabular data.
>
> **W2&A2: In designing the "valid bit mechanism," the authors focus on extrema values at the ends of the distribution, which is logical but may underutilize the central parts of the distribution. To validate this choice, theoretical proof or empirical evidence is needed to show that this approach outperforms one that also uses centered values for repeated information, as repetition might improve robustness. Could the authors provide theoretical or experimental comparisons between the "valid bit mechanism" and alternative approaches, such as information repetition, to clarify its effectiveness and superiority?**
>
> A: We appreciate your suggestion and have implemented a new bit accuracy calculation based on your idea. Specifically, we evaluated centered values (i.e., between $\Phi^{-1}(0.25)$ and $\Phi^{-1}(0.75)$). If the centered values in the first half match those in the second half, we assume the bits are accurate. The revised valid bit accuracy equation becomes:
>
>
> $A_{\text{cbit}} = \frac{\sum_{i=1}^{m/2} \mathbb{I}\left((d_i = 1 \text{ or } 2) \text{ and } (d_{m/2+i} = 1 \text{ or } 2)\right)}{\sum_{i=1}^{m/2} \mathbb{I}(d_i = 1 \text{ or } 2)}$
>
> We derived the central bit accuracy under Gaussian noise $\epsilon \sim N(0, \sigma)$), using the same setting at the end of Section 3. The resulting expected accuracy is:
>
> $$
> \mathbb{E}\left[A_{\text{cbit}}\right]=4\left(\int_{\Phi^{-1}(0.25)}^{\Phi^{-1}(0.75)}\left[\Phi\left(\frac{\Phi^{-1}(0.75) \sqrt{1+\sigma^2} - x}{\sigma}\right) - \Phi\left(\frac{\Phi^{-1}(0.25) \sqrt{1+\sigma^2} - x}{\sigma}\right)\right] \phi(x) dx\right)^2 + 16\left(\int_{\Phi^{-1}(0.75)}^{\infty}\left[\Phi\left(\frac{\Phi^{-1}(0.75) \sqrt{1+\sigma^2} - x}{\sigma}\right) - \Phi\left(\frac{\Phi^{-1}(0.25) \sqrt{1+\sigma^2} - x}{\sigma}\right)\right] \phi(x) dx\right)^2
> $$
>
>
> The following table summarizes the expected accuracy across three strategies: no valid bit mechanism, valid bit on tail values, and central bit on central values.
>
> | $\sigma$ | Expected acc w/o valid bit | Expected acc with valid bit | Expected acc with central bit |
> |------------|----------------------------|-------------------------------------|--------------------------------------|
> | 0          | 1.000                      | 1.000                               | 1.000                                |
> | 0.25       | 0.856                      | 0.980                               | 0.783                                |
> | 0.5        | 0.748                      | 0.909                               | 0.642                                |
> | 0.75       | 0.674                      | 0.817                               | 0.567                                |
> | 1          | 0.625                      | 0.740                               | 0.532                                |
>
> At equivalent noise levels, the valid bit mechanism focusing on tail values achieves higher bit accuracy and greater robustness compared to strategies relying on central values or random latent. The lower accuracy for central values suggests that focusing on the tails provides superior resistance to noise. The following figure (hosted anonymously) illustrates this comparison, in which the expected bit accuracy of the central bit mechanism has been added to Figure 2.
>
> [Figure A. Comparison of expected bit accuracy](https://postimg.cc/Dm2ryNXj)

---

> ### Author Response · Authors · 2024-11-18
> **Response to Reviewer CBoB (2)**
>
> **W3&A3 Given that the current experimental results show the proposed method’s performance is not consistently superior to previous watermarking methods, can the authors provide additional data or a more detailed explanation of these outcomes? How do they address this limitation to demonstrate the proposed method's advantages more clearly?**
>
> A: We acknowledge that the robustness of our method, as shown in Table 2, is not consistently superior to Gaussian Shading [1] across all datasets. However, we believe that our method achieves a better trade-off between robustness and data quality. Here is our reasoning:
>
> 1. Detectability in Terms of p-value: While the Z-scores in Table 2 for our method (Our*) are indeed lower, the detectability of our method remains competitive. Even in the worst-case scenario for our method, the Z-score (Diabetes dataset under 20% cell deletion attacks) corresponds to a p-value of 1.5e-4. This implies that even under the strongest attack (5k rows), our method maintains a low false positive rate of 1.5e-4. This shows that the detectability of our method is still reliable, despite its slightly lower robustness.
>
> 2. Trade-off with Data Quality: In contrast, Gaussian Shading, while exhibiting superior robustness, significantly compromises data quality. This is due to its reliance on using the same latent seed across all rows, which introduces noticeable patterns in the watermarked data. Our method avoids such performance drop, preserving the overall quality of the dataset better than Gaussian Shading.
>
> We recognize the importance of illustrating these trade-offs more effectively. To address this, we will include a new figure in our paper that highlights the trade-off between detectability and data quality across different watermarking methods. This figure will display the theoretical false positive rate (p-value) on the x-axis and the average of four different data quality metrics (from Table 1, specifically `Shape`, `Trend`, `Logistic`, `MLE`) on the y-axis, evaluated under the strongest attack settings. This visualization will underscore the advantages of our method in achieving a superior trade-off.
>
> We present such figures (hosted anonymously) in the following: the trade-off between p-value under various attacks and the average data quality. Notably, our method (Ours*, represented by filled plus markers) predominantly occupies the upper-left region, indicating superior performance in most scenarios, except under cell deletion attacks and Gaussian noise attacks on the Diabetes dataset. While GS demonstrates robust detectability, it consistently remains in the lower-left region of the figures, highlighting its trade-off of reduced data quality for robustness.
>
> [Figure B. Trade-off Analysis: Quality and Robustness Under 20% Row Deletion](https://postimg.cc/5X1YnYSP)
>
> [Figure C. Trade-off Analysis: Quality and Robustness Under 3-Column Deletion](https://postimg.cc/Yv34VrXt)
>
> [Figure D. Trade-off Analysis: Quality and Robustness Under 20% Cell Deletion](https://postimg.cc/QV2WKn9Z)
>
> [Figure E. Trade-off Analysis: Quality and Robustness Under 20% Gaussian Noise](https://postimg.cc/crftRMwg)
>
> ### References
> [1] Yang, Zijin, et al. "Gaussian Shading: Provable Performance-Lossless Image Watermarking for Diffusion Models." Proceedings of the IEEE/CVF Conference on Computer Vision and Pattern Recognition. 2024.
>
> [2] He, Hengzhi, et al. "Watermarking generative tabular data." arXiv preprint arXiv:2405.14018 (2024).

---

> > ### Comment · Reviewer_CBoB · 2024-11-25
> > **Raise my rating**
> >
> > Thanks to the authors for considering and  responding all the issues I concern.
> >
> >  In authors' response, they can positively reply all of my questions, and I can perceive a significant workload contained in the author's reply, this also demonstrates the sincerity and seriousness of the authors.
> >
> >  In detail, authors can demonstrate, both theoretically and experimentally, the soundness of the valid bit mechanism they use and its superiority compared to the alternative methods. For the experimental results that cannot be as good as the previous work, authors can give the explanation from the aspect of a better trade-off between robustness and data quality and promise to add corresponding figure in their paper for highlighting. Thus the only existing point is the incremental novelty, but considering authors' specific adaptations for tabular diffusion model, the situation that authors claim their paper to be the first paper to introduce watermarking during the sampling phase for tabular data, and authors' promise that in their future work they will further address the challenges caused by the unique properties of tabular data, which can add to the novelty of their work, I think this paper can be encouraged to inspire more works in this field, so as to arouse more improvements about this topic.
> >
> > Based on the above-mentioned consideration, I decide to increase the score of this paper from 5 to 6. I hope that in the next version of this paper, the authors will keep their promise and make the corresponding improvements they mentioned.

---

> > > ### Author Response · Authors · 2024-11-25
> > >
> > > Dear Reviewer CBoB,
> > >
> > > Thank you for your thorough and thoughtful review, as well as for recognizing the effort and sincerity we dedicated to addressing your concerns. We deeply appreciate your constructive feedback and your acknowledgment of our work.
> > >
> > > We assure you that we will uphold our promise and aim to complete the revisions before this period concludes.

---

### Official Review · Reviewer_7YMM · 2024-11-04

**Soundness:** 3
**Presentation:** 3
**Contribution:** 3
**Rating:** 8
**Confidence:** 3

**Summary:**

This paper introduces TABWAK, a novel approach to watermarking tabular generative models. To counter the application-specific attacks on tabular data, namely row shuffling, deletion, and recording, the author proposes a row-wise embedding method. To avoid loss of data diversity, the author proposed self-cloning and shuffling techniques in the latent space of the diffusion model. Through VAE and ddim inversion, the author can track the original latent used for the generation which will then be unshuffled to retrieve user identity.

**Strengths:**

This paper is well-written and easy to follow, the experiment results are thorough, and informative. The idea the author proposed in the paper is novel and interesting.

**Weaknesses:**

The attacks considered in this paper (e.g. row shuffling, deletion, etc) are somewhat basic and limited. The author could benefit from testing on more sophisticated attack methodologies for example adversarial attacks, where the attacker maximizes the distance in VAE or DDIM latent space while constraining the l2 norm of the perturbation to tabular data. However, I'm unfamiliar with the attack and misusage cases of tabular data, so a justification of the potential attack cases may also be helpful.

**Questions:**

1. I'm not an expert on watermarking tabular data and the field seems relatively new. In the introduction section, the author mentioned that "it is paramount to ensure its traceability and auditability to avoid harm and misusages" I wonder if the author could enlighten me on the potential cases of misusages and harm. I believe it will also be beneficial for the author to include this in the introduction for a broader audience.

2. Again, since I'm not an expert on tabular data generation, and this method relies on the dimensionality of latent space. What is the key capacity for this method? More specifically, what does m equal in your experiments? I assume this is a data and model-dependent hyperparameter that is fixed in your experiment since no retraining is needed, and it heavily affects the practicability of your method since it will affect the upper bound of your key capacity. A brief explanatory note on this could be beneficial.

---

> ### Author Response · Authors · 2024-11-18
> **Response to Reviewer 7YMM (1)**
>
> Thank you for your valuable comments.
>
> **W1 The attacks considered in this paper (e.g. row shuffling, deletion, etc) are somewhat basic and limited. The author could benefit from testing on more sophisticated attack methodologies for example adversarial attacks, where the attacker maximizes the distance in VAE or DDIM latent space while constraining the l2 norm of the perturbation to tabular data. However, I'm unfamiliar with the attack and misusage cases of tabular data, so a justification of the potential attack cases may also be helpful**
>
> A: Based on your and other reviewers' suggestions, we have implemented two new attacks designed to manipulate the latent space.  One post-processing watermarking method [1] is also incorporated into the evaluation. The first attack, referred to as the **Regeneration Attack**, is inspired by the approach in DiffPure[2]. In this attack, we employ the decoder inversion described in our paper to map the watermarked table to the latent representation, denoted as $\hat{\mathbf{z}}_0^W$. Subsequently, we use DDIM inversion to generate $\hat{\mathbf{z}}_T^W$. This newly generated initial latent representation is then used to reconstruct the tabular data. The regenerated tabular data is evaluated using the watermarking method. The results of the regeneration attack are presented in the table below.
>
> | Dataset  | TR   | GS    | Ours  | Ours\* | Post-processing |
> | -------- | ---- | ----- | ----- | ------ | --------------- |
> | Shoppers | 4.73 | 22.30 | 11.02 | 35.10  | 0.00            |
> | Magic    | 4.85 | 36.53 | 13.68 | 25.09  | 0.21            |
> | Adult    | 0.54 | 46.42 | 42.08 | 30.50  | 0.01            |
> | Credit   | 5.34 | 84.06 | 10.86 | 22.13  | 0.07            |
> | Diabetes | 6.29 | 55.76 | 5.82  | 7.04   | 0.03            |
>
> **Table A. Robustness of different watermarking methods against the regeneration attack: Average Z-score on 5K rows**
>
> From the results, we observe that during the regeneration process, the watermarks for sampling-based methods remain detectable (except for TR on the Adult dataset, where it fails even without the attack). In contrast, the watermark applied through post-processing is entirely removed during regeneration.
>
> The second attack, referred to as the **Embedding Attack**, is inspired by WAVES [3]. Utilizing our encoder $\mathcal{E}$, which maps the tabular data $(X_{num}, X_{cat})$ to a latent representation, we introduce perturbations to the numerical component of the tabular data, denoted as $X_{num}^{adv}$. These perturbations aim to deviate the latent representation of the adversarial table from that of the original watermarked table, $X_{num}$, while staying within a perturbation constraint. Formally, this is expressed as:
>
> $$
> \max \left\|\mathcal{E}(X_{num}^{adv}, X_{cat}) - \mathcal{E}(X_{num}, X_{cat})\right\|_2,
> $$
>
> subject to the constraint
>
> $$
> \left|X_{n u m}^{a d v}-X_{n u m}\right| \leq \epsilon \cdot\left|X_{n u m}\right|
> $$
>
> In our setting, $\epsilon$ is set to 0.2. The results are presented in the table below.
>
> | Dataset  | TR   | GS    | Ours | Ours\* | Post-processing |
> | -------- | ---- | ----- | ---- | ------ | --------------- |
> | shoppers | 0.31 | 22.49 | 0.00 | 28.69  | 0.00            |
> | magic    | 0.33 | 36.54 | 8.41 | 21.61  | 0.08            |
> | adult    | 0.00    | 56.29 | 0.08 | 27.00  | 0.06            |
> | credit   | 0.04 | 79.43 | 0.00 | 11.12  | 0.00            |
> | diabetes | 1.27 | 48.28 | 2.90 | 7.42   | 0.00            |
>
>
> **Table B. The Robustness of different watermarking methods Against Embedding Attack: Average Z-score on 5K rows**
>
> From the results, we observe that in this attack setting, our method with the valid bit mechanism (Ours*) and Gaussian Shading (GS) demonstrates strong robustness against attacks. In contrast, the post-processing watermark fails across all datasets, as the perturbation completely destroys the watermark.
>
> For the potential attack cases, please see the following answer to Q1.

---

> ### Author Response · Authors · 2024-11-18
> **Response to Reviewer 7YMM (2)**
>
> **Q1 In the introduction section, the author mentioned that "it is paramount to ensure its traceability and auditability to avoid harm and misusages" I wonder if the author could enlighten me on the potential cases of misusages and harm. I believe it will also be beneficial for the author to include this in the introduction for a broader audience.**
>
> A: Thanks for your suggestions. Synthetic tabular is the most common modality in industry and organizations, which increasingly embrace synthetic data as a privacy-preserving data sharing solution [4-6].  It is important for the synthetic data generator to verify if a piece of table is generated by itself and then take responsibility for the (misa)usage of such data. Synthetic tables pose subtler yet significant risks.
> For instance: 1) Financial Fraud: Synthetic datasets can manipulate performance metrics, enabling hedge funds to fabricate high returns and conceal losses. Watermarking ensures that only genuine data is used for informed decision-making. 2) Healthcare Misdiagnosis: Altered synthetic patient data can skew diagnostic tools or treatment recommendations, potentially leading to issues like over-prescription of medications. Watermarking safeguards data integrity, fostering trust in healthcare models. 3) Regulatory Evasion: Companies may exploit synthetic data to falsify compliance records, inflate profits, or create misleading sustainability reports.
>
> Watermarking confirms data authenticity, ensuring reliability in audits. The watermarking technique can also protect the copyright of generated tabular data for the model owner, ensuring that the data's ownership and intellectual property rights are safeguarded by the model itself.
>
> We will include this in our introduction.
>
> **Q2 What is the key capacity for this method? More specifically, what does m equal in your experiments? I assume this is a data and model-dependent hyperparameter that is fixed in your experiment since no retraining is needed, and it heavily affects the practicability of your method since it will affect the upper bound of your key capacity. A brief explanatory note on this could be beneficial.**
>
> A: Thank you for your notification. Yes, in our method, $m$ is a model- and data-related hyperparameter, calculated as the product of the token dimension and the number of columns in the table. This is because, in Tabsyn, the model converts each column into the latent space using the same token space. The first one is model-related, where we use the default setting in TabSyn, which is 4. The second component is data-related and depends on the tabular data's structure. We will include this explanation in our paper to provide additional clarity.
>
> ### References
> [1] He, Hengzhi, et al. "Watermarking generative tabular data." arXiv preprint arXiv:2405.14018 (2024).
>
> [2] Nie, Weili, et al. "Diffusion Models for Adversarial Purification." International Conference on Machine Learning. PMLR, 2022.
>
> [3] An, Bang, et al. "Benchmarking the robustness of image watermarks." arXiv preprint arXiv:2401.08573 (2024).
>
> [4] Liu, Fan, et al. "Privacy-preserving synthetic data generation for recommendation systems." Proceedings of the 45th International
> ACM SIGIR Conference on Research and Development in Information Retrieval. 2022.
>
> [5] Qian, Zhaozhi, et al. "Synthetic data for privacy-preserving clinical risk prediction." Scientific Reports 14.1 (2024): 25676.
>
> [6] Potluru, Vamsi K., et al. "Synthetic data applications in finance." arXiv preprint arXiv:2401.00081 (2023).

---

> ### Author Response · Authors · 2024-11-24
> **Response to Reviewer 7YMM (3) (Supplement to W1)**
>
> Based on Reviewer hNPH's suggestions, we also developed an adaptive attack that focuses on the tail values in the original latent. We additionally introduce perturbations to the numerical values in the table, denoted as $X_{\text{num}}$. Specifically, we use the encoder to approximate the latent $\hat{z}_0$, then employ the diffusion model for DDIM inversion to estimate the initial latent $\hat{z}_T$. Our attack minimizes the tail values of the latent $\hat{z}_T$, which can be formally expressed as:
>
> $$
> \min_{X_{\mathrm{num}}^{\mathrm{adv}}} \left\| M_{\mathrm{tail}} \cdot \hat{z}_T \right\|_2,
> $$
>
> subject to the constraint:
>
> $$
> \left| X_{\mathrm{num}}^{\mathrm{adv}} - X_{\mathrm{num}} \right| \leq \epsilon \cdot \left| X_{\mathrm{num}} \right|,
> $$
>
> where
>
> $$
> \hat{z}_T = DDIM^{-1}(\mathcal{E}(X_{\mathrm{num}}^{\mathrm{adv}})),
> $$
>
> and
>
> $$
> M_{\mathrm{tail}}[i] =
> \begin{cases}
> 1 & \text{if } \hat{z}_T[i] < Q_{0.25}(\hat{z}_T) \text{ or } \hat{z}_T[i] > Q_{0.75}(\hat{z}_T), \\
> 0 & \text{otherwise.}
> \end{cases}
> $$
>
>
> In our experiments, $\epsilon$ is set to 0.2. For DDIM inversion, we limit the number of steps to 10 to accelerate the process and reduce backpropagation overhead during optimization. The final results are summarized below:
>
> | Dataset  | W/O Attack   | Embedding Attack    | Adaptive Attack |
> | -------- | ------------ | ------------------- | ---------------- |
> | Shoppers | 34.52        | 28.69              | 24.61            |
> | Magic    | 25.30        | 21.61              | 7.43             |
> | Adult    | 28.45        | 27.00              | 26.03            |
> | Credit   | 22.91        | 11.12              | 14.48            |
> | Diabetes | 7.86         | 7.42               | 2.15             |
>
> **Table C. The Robustness of TabWak Against Embedding and Adaptive Attacks: Average Z-score on 5K rows**
>
> From the results, we observe that, under the same $\epsilon$, adaptive attacks reduced the average Z-score more significantly in 4 out of 5 datasets. Notably, the attack was particularly successful on the `magic` and `diabetes` datasets. We attribute this to the fact that we only perturbed numerical columns in the tabular data. In these two datasets, all columns are numerical except for the target columns, while the other three datasets contain more categorical columns. We will include these additional results in the appendix.

---

> > ### Comment · Reviewer_7YMM · 2024-11-25
> >
> > I appreciate the authors for their detailed response and experiment results for additional attacks. All my questions and concerns have been well addressed. Although the methodology still has its weaknesses, e.g. limited key capacity, and robustness against certain learning-based attacks, I think this paper is insightful for future works in the field of tabular data watermarking, the presentation is complete and thorough. So I have raised my score accordingly.

---

> > > ### Author Response · Authors · 2024-11-25
> > >
> > > Dear Reviewer 7YMM,
> > >
> > > We are thrilled that our response has addressed your concerns. Your constructive feedback has been invaluable in refining our work, and we sincerely thank you for your thoughtful and thorough review.

---

### Official Review · Reviewer_w73M · 2024-11-04

**Soundness:** 3
**Presentation:** 3
**Contribution:** 3
**Rating:** 8
**Confidence:** 3

**Summary:**

The paper proposes the first sampling-phase watermarking method for tabular diffusion models. It also proposes a valid bit mechanism to enhance the robustness. Theoretical guarantee is provided for row-level diversity and detection. Extensive experiments show the effectiveness of the method.

**Strengths:**

1. The paper proposes the first sampling-phase watermarking method for tabular diffusion models.

2. To enhance the robustness, the paper proposes a valid bit mechanism.

3. The paper shows theoretical guarantee for the proposed method.

4. Extensive experiments validate the effectivenss and robustness of the proposed method.

**Weaknesses:**

There is one main concern.

Will the purification methods in image-domain be effective for the watermark in tabular diffusion models? For example, if DiffPure is used to purify the latent, will the watermark still be effective?

**Questions:**

Please see the Weaknesses part.

---

> ### Author Response · Authors · 2024-11-18
> **Response to the Reviewer w73M**
>
> Thank you for your valuable comments and positive feedback.
> In response to your questions, we have implemented two new attacks designed to manipulate the latent space. One post-processing watermarking method [1] is also incorporated into the evaluation. The first attack, referred to as the **Regeneration Attack**, is inspired by the approach in DiffPure[2]. In this attack, we employ the decoder inversion described in our paper to map the watermarked table to the latent representation, denoted as $\hat{\mathbf{z}}_0^W$. Subsequently, we use DDIM inversion to generate $\hat{\mathbf{z}}_T^W$. This newly generated initial latent representation is then used to reconstruct the tabular data. The regenerated tabular data is evaluated using the watermarking method. The results of the regeneration attack are presented in the table below.
>
> | Dataset  | TR   | GS    | Ours  | Ours\* | Post-processing |
> | -------- | ---- | ----- | ----- | ------ | --------------- |
> | Shoppers | 4.73 | 22.30 | 11.02 | 35.10  | 0.00            |
> | Magic    | 4.85 | 36.53 | 13.68 | 25.09  | 0.21            |
> | Adult    | 0.54 | 46.42 | 42.08 | 30.50  | 0.01            |
> | Credit   | 5.34 | 84.06 | 10.86 | 22.13  | 0.07            |
> | Diabetes | 6.29 | 55.76 | 5.82  | 7.04   | 0.03            |
>
> **Table A. Robustness of different watermarking methods against the regeneration attack: Average Z-score on 5K rows**
>
> From the results, we observe that during the regeneration process, the watermarks for sampling-based methods remain detectable (except for TR on the Adult dataset, where it fails even without the attack). In contrast, the watermark applied through post-processing is entirely removed during regeneration.
>
> The second attack, referred to as the **Embedding Attack**, is inspired by WAVES [3]. Utilizing our encoder $\mathcal{E}$, which maps the tabular data $(X_{num}, X_{cat})$ to a latent representation, we introduce perturbations to the numerical component of the tabular data, denoted as $X_{num}^{adv}$. These perturbations aim to deviate the latent representation of the adversarial table from that of the original watermarked table, $X_{num}$, while staying within a perturbation constraint. Formally, this is expressed as:
>
> $$
> \max \left\|\mathcal{E}(X_{num}^{adv}, X_{cat}) - \mathcal{E}(X_{num}, X_{cat})\right\|_2,
> $$
>
> subject to the constraint
>
> $$
> \left|X_{n u m}^{a d v}-X_{n u m}\right| \leq \epsilon \cdot\left|X_{n u m}\right|
> $$
>
> In our setting, $\epsilon$ is set to 0.2. The results are presented in the table below.
>
> | Dataset  | TR   | GS    | Ours | Ours\* | Post-processing |
> | -------- | ---- | ----- | ---- | ------ | --------------- |
> | shoppers | 0.31 | 22.49 | 0.00 | 28.69  | 0.00            |
> | magic    | 0.33 | 36.54 | 8.41 | 21.61  | 0.08            |
> | adult    | 0.00    | 56.29 | 0.08 | 27.00  | 0.06            |
> | credit   | 0.04 | 79.43 | 0.00 | 11.12  | 0.00            |
> | diabetes | 1.27 | 48.28 | 2.90 | 7.42   | 0.00            |
>
>
> **Table B. The Robustness of different watermarking methods Against Embedding Attack: Average Z-score on 5K rows**
>
> From the results, we observe that in this attack setting, our method with the valid bit mechanism (Ours*) and Gaussian Shading (GS) demonstrate strong robustness against attacks. In contrast, the post-processing watermark fails across all datasets, as the perturbation completely destroys the watermark.
>
> ### References
> [1] He, Hengzhi, et al. "Watermarking generative tabular data." arXiv preprint arXiv:2405.14018 (2024).
>
> [2] Nie, Weili, et al. "Diffusion Models for Adversarial Purification." International Conference on Machine Learning. PMLR, 2022.
>
> [3] An, Bang, et al. "Benchmarking the robustness of image watermarks." arXiv preprint arXiv:2401.08573 (2024).

---

### Author Response · Authors · 2024-11-24
**General Rebuttal**

**Dear Reviewers, ACs, and PCs,**

Thank you for your dedication, support, and insightful feedback. We deeply appreciate your suggestions, which have greatly enhanced our work. Below is a summary of the key updates and improvements we have made:

- Implemented post-processing watermark [1]. Compared results for watermark robustness (Reviewers jg7M, hNPH)
- Designed a new central-bit algorithm based on the central part of the distribution. Provided theoretical derivation on the expected bit accuracy. (Reviewer CBoB)
- Designed, implemented and evaluated **regeneration**, **embedding**  and **adptive** attacks (Reviewers w73M, 7YMM, hNPH)
- Provided additional figures on the tradeoff between the robustness and generation quality of different methods (Reviewers CBoB, jg7M)
- Motivate usefulness for watermarking tables via examples (Reviewers 7YMM, jg7M)
- Tested different setting of  hyperparameter _l_ (Reviewer hNPH)
- Clarified the role of the hyperparameter _m_ (Reviewer 7YMM)
- Clarified challenges and novelty of watermarking tabular data (Reviewer CBoB)
- Discussed the possibility and advantage when extending our method to the image domain (Reviewer jg7M)
- Discussed privacy and compatibility with different diffusion models (Reviewer hNPH)

**Best regards,**

The Authors


**References**

[1] He, Hengzhi, et al. "Watermarking generative tabular data." arXiv preprint arXiv:2405.14018 (2024).

---

> ### Author Response · Authors · 2024-11-28
> **Revisions**
>
> Based on the reviewers' suggestions, we have made the following revisions to our manuscript, with all changes highlighted in blue:
>
> - Added examples to motivate watermarking tables in the Introduction (Reviewers 7YMM, jg7M).
> - Clarified hyperparameter m in Section 3 (Reviewer 7YMM).
> - Included figures on robustness vs. generation quality trade-offs in Figure 3 (Reviewers CBoB, jg7M).
> - Compared post-processing watermark robustness in Appendix F.3 (Reviewers jg7M, hNPH).
> - Evaluated regeneration, embedding, and adaptive attacks in Appendix F.4 (Reviewers w73M, 7YMM, hNPH).
> - Explored hyperparameter l settings in Appendix F.5 (Reviewer hNPH).
> - Fixed typos in Figures 4 and 5.
>
> We sincerely thank the reviewers for their valuable suggestions and efforts to improve our manuscript.

---

### Meta-Review · Area_Chair_fgLM · 2024-12-11

**Metareview:**

The paper proposes a novel method to watermark the tabular generative models called TABWAK. In this paper, the authors clearly illustrate the unique properties of the tabular diffusion models compared with other modalities and then propose many techniques for better watermarking, including row-wise embedding, self-cloning, and shuffling techniques. Furthermore, the authors also provide comprehensive empirical evaluations and theoretical analysis of their method to demonstrate their effective and robust performance. During the rebuttal period, the authors and reviewers had an active discussion on the problem's importance, method robustness, ablation studies, etc. These discussions, additional results, and revisions also strengthen the paper. Therefore, all the reviewers and I agree this paper can be accepted by ICLR.

Strengths:
1. The paper is clear, and their discussions of the difference between tabular DMs and other DMs watermarks are good and insightful.

2. Comprehensive experiments have been conducted to demonstrate their TabWak's effectiveness and robustness.

3. The theoretical analysis also guarantees their methods' effectiveness.

4. This paper is the first watermarking method to embed invisible signatures for tabular DMs. As tabular data is one of the widely used data types in practice, this work can have great impacts on both the research and social community.

Weaknesses:
The proposed methods are mainly based on former DM's watermarking methods. Therefore, the technical novelty is not strong.

In summary, this paper is informative and the topic they studied is important and new. Therefore, I tend to accept this paper as a spotlight paper.

**Additional Comments On Reviewer Discussion:**

The reviewers and authors have an active discussion in the rebuttal period. After the rebuttal, the papers' evaluation is more comprehensive and the topics' importance is more clear.

---

### Decision · Program_Chairs · 2025-01-22

Accept (Spotlight)